# Maximum Entropy Reinforcement Learning via Energy-Based Normalizing Flow

**Chen-Hao Chao**[*1,2]     **Chien Feng**[*1]     **Wei-Fang Sun**[2]

**Cheng-Kuang Lee**[2]     **Simon See**[2]     **Chun-Yi Lee**[†1]

[1] Elsa Lab, National Tsing Hua University, Hsinchu City, Taiwan
[2] NVIDIA AI Technology Center, NVIDIA Corporation, Santa Clara, CA, USA

## Abstract

Existing Maximum-Entropy (MaxEnt) Reinforcement Learning (RL) methods for continuous action spaces are typically formulated based on actor-critic frameworks and optimized through alternating steps of policy evaluation and policy improvement. In the policy evaluation steps, the critic is updated to capture the soft Q-function. In the policy improvement steps, the actor is adjusted in accordance with the updated soft Q-function. In this paper, we introduce a new MaxEnt RL framework modeled using Energy-Based Normalizing Flows (EBFlow). This framework integrates the policy evaluation steps and the policy improvement steps, resulting in a single objective training process. Our method enables the calculation of the soft value function used in the policy evaluation target without Monte Carlo approximation. Moreover, this design supports the modeling of multi-modal action distributions while facilitating efficient action sampling. To evaluate the performance of our method, we conducted experiments on the MuJoCo benchmark suite and a number of high-dimensional robotic tasks simulated by Omniverse Isaac Gym. The evaluation results demonstrate that our method achieves superior performance compared to widely-adopted representative baselines.

## 1 Introduction

Maximum-Entropy (MaxEnt) Reinforcement Learning (RL) [1–17] has emerged as a prominent method for modeling stochastic policies. Different from standard RL, MaxEnt RL integrates the entropy of policies as rewards, which leads to a balanced exploration-exploitation trade-off during training. This approach has demonstrated improved robustness both theoretically and empirically [17–19]. Building on this foundation, many studies leveraging MaxEnt RL have shown superior performance on continuous-control benchmark environments [8, 9] and real-world applications [20–22].

An active research domain in MaxEnt RL concentrates on the learning of the soft Q-function [8–15]. These methods follow the paradigm introduced in soft Q-learning (SQL) [8]. They parameterize the soft Q-function as an energy-based model [23] and optimize it based on the soft Bellman error [8] calculated from rewards and the soft value function. However, this approach presents two challenges. First, sampling from an energy-based model requires a costly Monte Carlo Markov Chain (MCMC) [24, 25] or variational inference [26] process, which can result in inefficient interactions with environments. Second, the calculation of the soft value function can involve computationally

---

[*]Equal contribution.
[†]Corresponding author. Email: `cylee@cs.nthu.edu.tw`

38th Conference on Neural Information Processing Systems (NeurIPS 2024).

infeasible integration, which requires an effective approximation method. To tackle these issues, various methods [8–15] were proposed, all grounded in a common design philosophy. To address the first challenge, these methods suggest operating on an actor-critic framework and optimizing it through alternating steps of policy evaluation and policy improvement. For the second challenge, they resort to Monte Carlo methods to approximate the soft value function using sets of random samples. Although these two issues can be circumvented, these methods still have their drawbacks. The actor-critic design introduces an additional optimization process for training the actor, which may lead to optimization errors in practice [27]. Moreover, the results of Monte Carlo approximation may be susceptible to estimation errors and variances when there are an insufficient number of samples [28–30].

Instead of using energy-based models to represent MaxEnt RL frameworks, this paper investigates an alternative method employing normalizing flows (i.e., flow-based models), which offer solutions to the aforementioned challenges. Our framework is inspired by the recently introduced Energy-Based Normalizing Flows (EBFlow) [31]. This design facilitates the derivation of an energy function from a flow-based model while supporting efficient sampling, which enables a unified representation of both the soft Q-function and its corresponding action sampling process. This feature allows the integration of the policy evaluation and policy improvement steps into a single objective training process. In addition, the probability density functions (pdf) of flow-based models can be calculated efficiently without approximation. This characteristic permits the derivation of an exact representation for the soft value function. Our experimental results demonstrate that the proposed framework exhibits superior performance on the commonly adopted MuJoCo benchmark [32, 33]. Furthermore, the evaluation results on the Omniverse Isaac Gym environments [34] indicate that our framework excels in performing challenging robotic tasks that simulate real-world scenarios.

## 2 Background and Related Works

In this section, we walk through the background material and the related works. We introduce the objective of MaxEnt RL in Section 2.1, describe existing actor-critic frameworks and soft value estimation methods in Section 2.2, and elaborate on the formulation of Energy-Based Normalizing Flow (EBFlow) in Section 2.3.

### 2.1 Maximum Entropy Reinforcement Learning

In this paper, we consider a Markov Decision Process (MDP) defined as a tuple $(\mathcal{S}, \mathcal{A}, p_T, \mathcal{R}, \gamma, p_0)$, where $\mathcal{S}$ is a continuous state space, $\mathcal{A}$ is a continuous action space, $p_T : \mathcal{S} \times \mathcal{S} \times \mathcal{A} \to \mathbb{R}_{\geq 0}$ is the pdf of a next state $\mathbf{s}_{t+1}$ given a current state $\mathbf{s}_t$ and a current action $\mathbf{a}_t$ at timestep $t$, $\mathcal{R} : \mathcal{S} \times \mathcal{A} \to \mathbb{R}$ is the reward function, $0 < \gamma < 1$ is the discount factor, and $p_0$ is the pdf of the initial state $\mathbf{s}_0$. We adopt $r_t$ to denote $\mathcal{R}(\mathbf{s}_t, \mathbf{a}_t)$, and use $\rho_\pi(\mathbf{s}_t, \mathbf{a}_t)$ to represent the state-action marginals of the trajectory distribution induced by a policy $\pi(\mathbf{a}_t|\mathbf{s}_t)$ [8].

Standard RL defines the objective as $\pi^* = \text{argmax}_\pi \sum_t \mathbb{E}_{(\mathbf{s}_t,\mathbf{a}_t)\sim\rho_\pi}[r_t]$ and has at least one deterministic optimal policy [35, 36]. In contrast, MaxEnt RL [4] augments the standard RL objective with the entropy of a policy at each visited state $\mathbf{s}_t$. The objective of MaxEnt RL is written as follows:

$$\pi^*_{\text{MaxEnt}} = \underset{\pi}{\text{argmax}} \sum_t \mathbb{E}_{(\mathbf{s}_t,\mathbf{a}_t)\sim\rho_\pi} \left[ r_t + \alpha \mathcal{H}(\pi(\,\cdot\,|\mathbf{s}_t)) \right], \tag{1}$$

where $\mathcal{H}(\pi(\,\cdot\,|\mathbf{s}_t)) \triangleq \mathbb{E}_{\mathbf{a}\sim\pi(\cdot|\mathbf{s}_t)}[-\log\pi(\mathbf{a}|\mathbf{s}_t)]$ and $\alpha \in \mathbb{R}_{>0}$ is a temperature parameter for determining the relative importance of the entropy term against the reward. An extension of Eq. (1) defined with $\gamma$ is discussed in [8]. To obtain $\pi^*_{\text{MaxEnt}}$ described in Eq. (1), the authors in [8] proposed to minimize the soft Bellman error for all states and actions. The solution can be expressed using the optimal soft Q-function $Q^*_{\text{soft}} : \mathcal{S} \times \mathcal{A} \to \mathbb{R}$ and soft value function $V^*_{\text{soft}} : \mathcal{S} \to \mathbb{R}$ as follows:

$$\pi^*_{\text{MaxEnt}}(\mathbf{a}_t|\mathbf{s}_t) = \exp\left( \frac{1}{\alpha}(Q^*_{\text{soft}}(\mathbf{s}_t, \mathbf{a}_t) - V^*_{\text{soft}}(\mathbf{s}_t)) \right), \text{ where} \tag{2}$$

$$Q^*_{\text{soft}}(\mathbf{s}_t, \mathbf{a}_t) = r_t + \gamma \mathbb{E}_{\mathbf{s}_{t+1}\sim p_T} \left[ V^*_{\text{soft}}(\mathbf{s}_{t+1}) \right], \quad V^*_{\text{soft}}(\mathbf{s}_t) = \alpha \log \int \exp\left( \frac{1}{\alpha} Q^*_{\text{soft}}(\mathbf{s}_t, \mathbf{a}) \right) d\mathbf{a}. \tag{3}$$

In practice, a policy can be modeled as $\pi_\theta(\mathbf{a}_t|\mathbf{s}_t) = \exp(\frac{1}{\alpha}(Q_\theta(\mathbf{s}_t, \mathbf{a}_t) - V_\theta(\mathbf{s}_t)))$ with parameter $\theta$, where the soft Q-function and the soft value function are expressed as $Q_\theta(\mathbf{s}_t, \mathbf{a}_t)$ and $V_\theta(\mathbf{s}_t) =$

$\alpha \log \int \exp \left( \frac{1}{\alpha} Q_\theta(\mathbf{s}_t, \mathbf{a}_t) \right) d\mathbf{a}_t$, respectively. Given an experience reply buffer $\mathcal{D}$ that stores transition tuples $(\mathbf{s}_t, \mathbf{a}_t, r_t, \mathbf{s}_{t+1})$, the training objective of $Q_\theta$ (which can then be used to derive $V_\theta$ and $\pi_\theta$) can be written as the following equation according to the soft Bellman errors:

$$\mathcal{L}(\theta) = \mathbb{E}_{(\mathbf{s}_t, \mathbf{a}_t, r_t, \mathbf{s}_{t+1}) \sim \mathcal{D}} \left[ \frac{1}{2} \left( Q_\theta(\mathbf{s}_t, \mathbf{a}_t) - (r_t + \gamma V_\theta(\mathbf{s}_{t+1})) \right)^2 \right]. \tag{4}$$

Nonetheless, directly using the objective in Eq. (4) presents challenges for two reasons. First, drawing samples from an energy-based model (i.e., $\pi_\theta(\mathbf{a}_t|\mathbf{s}_t) \propto \exp(Q_\theta(\mathbf{s}_t, \mathbf{a}_t)/\alpha)$) requires a costly MCMC or variational inference process [26, 37], which makes the interaction with the environment inefficient. Second, the calculation of the soft value function involves integration, which require stochastic approximation methods [28–30] to accomplish. To address these issues, the previous MaxEnt RL methods [8–14] adopted actor-critic frameworks and introduced a number of techniques to estimate the soft value function. These methods are discussed in the next subsection.

## 2.2 Actor-Critic Frameworks and Soft Value Estimation in MaxEnt RL

Previous MaxEnt RL methods [8–15] employed actor-critic frameworks, in which the critic aims to capture the soft Q-function, while the actor learns to sample actions based on this soft Q-function. Available choices for modeling the actor include Gaussian models [9], Gaussian mixture models [38], variational autoencoders (VAE) [15, 13, 39], normalizing flows [10, 11], and amortized SVGD (A-SVGD) [8, 40], all of which support efficient sampling. The separation of the actor and the critic prevents the need for costly MCMC processes during sampling. However, this design induces additional training steps aimed to minimize the discrepancy between them. Let $\pi_\theta(\mathbf{a}_t|\mathbf{s}_t) \propto \exp(\frac{1}{\alpha} Q_\theta(\mathbf{s}_t, \mathbf{a}_t))$ and $\pi_\phi(\mathbf{a}_t|\mathbf{s}_t)$ denote the pdfs defined through the critic and the actor, respectively. The objective of this additional training process is formulated according to the reverse KL divergence $\mathbb{D}_{\mathrm{KL}}[\pi_\phi(\cdot|\mathbf{s}_t)||\pi_\theta(\cdot|\mathbf{s}_t)]$ between $\pi_\phi$ and $\pi_\theta$, and is typically reduced as follows [9]:

$$\mathcal{L}(\phi) = \mathbb{E}_{\mathbf{s}_t \sim \mathcal{D}} \left[ -\mathbb{E}_{\mathbf{a}_t \sim \pi_\phi}[Q_\theta(\mathbf{s}_t, \mathbf{a}_t) - \alpha \log \pi_\phi(\mathbf{a}_t|\mathbf{s}_t)] \right]. \tag{5}$$

The optimization processes defined by the objective functions $\mathcal{L}(\theta)$ and $\mathcal{L}(\phi)$ in Eqs. (4) and (5) are known as the policy evaluation steps and policy improvement steps [9], respectively. Through alternating updates according to $\nabla_\theta \mathcal{L}(\theta)$ and $\nabla_\phi \mathcal{L}(\phi)$, the critic learns directly from the reward signals to estimate the soft Q-function, while the actor learns to draw samples based on the distribution defined by the critic.

Although the introduction of the actor enhances sampling efficiency, calculating the soft value function in Eq. (3) still requires Monte Carlo approximations for the computationally infeasible integration operation. Prior soft value estimation methods can be categorized into two groups: soft value estimation in Soft Q-Learning (SQL) and that in Soft Actor-Critic (SAC), with the former yielding a larger estimate than the latter, derived from Jensen's inequality (i.e., Proposition A.1 in Appendix A.1). These two soft value estimation methods are discussed in the following paragraphs.

**Soft Value Estimation in SQL.** Soft Q-Learning [8] leverages importance sampling to convert the integration in Eq. (3) into an expectation, which can be estimated using a set of independent and identically distributed (i.i.d.) samples. To ensure the estimation variance is small, the authors in [8] proposed to utilize samples drawn from $\pi_\phi$. Let $\{\mathbf{a}^{(i)}\}_{i=1}^M$ be a set of $M$ samples drawn from $\pi_\phi$. The soft value function is approximated based on the following formula:

$$V_\theta(\mathbf{s}_t) = \alpha \log \int \exp\left( Q_\theta(\mathbf{s}_t, \mathbf{a})/\alpha \right) d\mathbf{a} = \alpha \log \int \pi_\phi(\mathbf{a}|\mathbf{s}_t) \frac{\exp\left( Q_\theta(\mathbf{s}_t, \mathbf{a})/\alpha \right)}{\pi_\phi(\mathbf{a}|\mathbf{s}_t)} d\mathbf{a}$$
$$= \alpha \log \mathbb{E}_{\mathbf{a} \sim \pi_\phi} \left[ \frac{\exp\left( Q_\theta(\mathbf{s}_t, \mathbf{a})/\alpha \right)}{\pi_\phi(\mathbf{a}|\mathbf{s}_t)} \right] \approx \alpha \log \left( \frac{1}{M} \sum_{i=1}^M \frac{\exp\left( Q_\theta(\mathbf{s}_t, \mathbf{a}^{(i)})/\alpha \right)}{\pi_\phi(\mathbf{a}^{(i)}|\mathbf{s}_t)} \right). \tag{6}$$

Eq. (6) has the least variance when $\pi_\phi(\cdot|\mathbf{s}_t) \propto \exp(Q_\theta(\mathbf{s}_t, \cdot)/\alpha)$ [29]. In addition, as $M \to \infty$, the law of large numbers ensures that this estimation converge to $V_\theta(\mathbf{s}_t)$ [41].

**Soft Value Estimation in SAC.** Soft Actor-Critic [9] and its variants [10, 11, 13, 14, 12] reformulated the soft value function $V_\theta(\mathbf{s}_t) = \alpha \log \int \exp\left( Q_\theta(\mathbf{s}_t, \mathbf{a})/\alpha \right) d\mathbf{a}$ as its equivalent form $\mathbb{E}_{\mathbf{a} \sim \pi_\theta}[Q_\theta(\mathbf{s}_t, \mathbf{a}) - \alpha \log \pi_\theta(\mathbf{a}|\mathbf{s}_t)]$ based on the relationship that $\pi_\theta(\mathbf{a}|\mathbf{s}_t) = \exp(\frac{1}{\alpha}(Q_\theta(\mathbf{s}_t, \mathbf{a})) - $

$V_\theta(\mathbf{s}_t)$). By assuming that the policy improvement loss $\mathcal{L}(\phi)$ is small (i.e., $\pi_\theta \approx \pi_\phi$), the soft value function $V_\theta$ can be estimated as follows:

$$V_\theta(\mathbf{s}_t) = \mathbb{E}_{\mathbf{a} \sim \pi_\theta}[Q_\theta(\mathbf{s}_t, \mathbf{a}) - \alpha \log \pi_\theta(\mathbf{a}|\mathbf{s}_t)]$$

$$\approx \mathbb{E}_{\mathbf{a} \sim \pi_\phi}[Q_\theta(\mathbf{s}_t, \mathbf{a}) - \alpha \log \pi_\phi(\mathbf{a}|\mathbf{s}_t)] \approx \frac{1}{M} \sum_{i=1}^{M} \left( Q_\theta(\mathbf{s}_t, \mathbf{a}^{(i)}) - \alpha \log \pi_\phi(\mathbf{a}^{(i)}|\mathbf{s}_t) \right). \tag{7}$$

An inherent drawback of the estimation in Eq. (7) is its reliance on the assumption $\pi_\phi \approx \pi_\theta$. In addition, the second approximation involves Monte Carlo estimation with $M$ samples $\{\mathbf{a}^{(i)}\}_{i=1}^{M}$, where the computational cost increases with the number of samples $M$.

### 2.3 Energy-Based Normalizing Flows

Normalizing flows (i.e., flow-based models) are universal representations for pdf [42]. Given input data $\mathbf{x} \in \mathbb{R}^D$, a latent variable $\mathbf{z} \in \mathbb{R}^D$ with prior pdf $p_\mathbf{z}$, and an invertible function $g_\theta = g_\theta^L \circ \cdots \circ g_\theta^1$ modeled as a neural network with $L$ layers, where $g_\theta^i : \mathbb{R}^D \to \mathbb{R}^D$, $\forall i \in \{1, \cdots, L\}$. According to the change of variable theorem and the distributive property of the determinant operation, a parameterized pdf $p_\theta$ can be described as follows:

$$p_\theta(\mathbf{x}) = p_\mathbf{z}\left(g_\theta(\mathbf{x})\right) \prod_{i=1}^{L} \left| \det\left(\mathbf{J}_{g_\theta^i}(\mathbf{x}^{i-1})\right) \right|, \tag{8}$$

where $\mathbf{x}^0 \triangleq \mathbf{x}$ is the input, $\mathbf{x}^i = g_\theta^i \circ \cdots \circ g_\theta^1(\mathbf{x})$ is the output of the $i$-th layer, and $\mathbf{J}_{g_\theta^i}(\mathbf{x}^{i-1}) \triangleq \frac{\partial}{\partial \mathbf{x}^{i-1}} g_\theta^i(\mathbf{x}^{i-1})$ represents the Jacobian of the $i$-th layer of $g_\theta$ with respect to $\mathbf{x}^{i-1}$. To draw samples from $p_\theta$, one can first sample $\mathbf{z}$ from $p_\mathbf{z}$ and then derive $g_\theta^{-1}(\mathbf{z})$. To facilitate efficient computation of the pdf and the inverse of $g_\theta$, one can adopt existing architectural designs [43–48] for $g_\theta$. Popular examples involve autoregressive layers [43–45] and coupling layers [46–48], which utilizes specially designed architectures to speed up the calculation.

Energy-Based Normalizing Flows (EBFlow) [31] were recently introduced to reinterpret flow-based models as energy-based models. In contrast to traditional normalizing flow research [46, 47, 49, 50] that focuses on the use of effective non-linearities, EBFlow emphasizes the use of both linear and non-linear transformations in the invertible transformation $g_\theta$. Such a concept was inspired by the development of normalizing flows with convolution layers [48, 51–54] or fully-connected layers [55, 56], linear independent component analysis (ICA) models [57, 58], as well as energy-based training techniques [58–60]. Let $\mathcal{S}_l = \{i \mid g_\theta^i \text{ is linear}\}$ and $\mathcal{S}_n = \{i \mid g_\theta^i \text{ is non-linear}\}$ represent the sets of indices of the linear and non-linear transformations in $g_\theta$, respectively. As shown in [31], the Jacobian determinant product in Eq. (8) can be decomposed according to $\mathcal{S}_n$ and $\mathcal{S}_l$. This decomposition allows a flow-based model to be reinterpreted as an energy-based model, as illustrated in the following equation:

$$p_\theta(\mathbf{x}) = p_\mathbf{z}\left(g_\theta(\mathbf{x})\right) \underbrace{\prod_{i \in \mathcal{S}_n} \left| \det\left(\mathbf{J}_{g_\theta^i}(\mathbf{x}^{i-1})\right) \right|}_{\text{Unnormalized Density}} \underbrace{\prod_{i \in \mathcal{S}_l} \left| \det\left(\mathbf{J}_{g_\theta^i}\right) \right|}_{\text{Const.}} \triangleq \underbrace{\exp\left(-E_\theta(\mathbf{x})\right)}_{\text{Unnormalized Density}} \underbrace{Z_\theta^{-1}}_{\text{Const.}}. \tag{9}$$

In EBFlow, the energy function $E_\theta(\mathbf{x})$ is defined as $-\log(p_\mathbf{z}\left(g_\theta(\mathbf{x})\right) \prod_{i \in \mathcal{S}_n} |\det(\mathbf{J}_{g_\theta^i}(\mathbf{x}^{i-1}))|)$ and the normalizing constant $Z_\theta = \int \exp(-E_\theta(\mathbf{x}))d\mathbf{x} = \prod_{i \in \mathcal{S}_l} |\det \mathbf{J}_{g_\theta^i}|^{-1}$ is independent of $\mathbf{x}$. The input-independence of $Z_\theta$ holds since $g_\theta^i$ is either a first-degree or zero-degree polynomial for any $i \in \mathcal{S}_l$, and thus its Jacobian is a constant to $\mathbf{x}^{i-1}$. This technique was originally proposed to reduce the training cost of maximum likelihood estimation for normalizing flows. However, we discovered that EBFlow is ideal for MaxEnt RL. Its unique capability to represent a parametric energy function with an associated sampler $g_\theta^{-1}$, and to calculate a normalizing constant $Z_\theta$ without integration are able to address the challenges mentioned in Section 2.2. We discuss our insights in the next section.

## 3 Methodology

In this section, we introduce our proposed MaxEnt RL framework modeled using EBFlow. In Section 3.1, we describe the formulation and discuss its training and inference processes. In Section 3.2,

we present two techniques for improving the training of our framework. Ultimately, in Section 3.3, we offer an algorithm summary.

## 3.1 MaxEnt RL via EBFlow

We propose a new framework for modeling **M**ax**E**nt RL using EBFl**ow**, which we call **MEow**. This framework possesses several unique features. First, as EBFlow enables simultaneous modeling of an unnormalized density and its sampler, MEow can unify the actor and the critic previously separated in MaxEnt RL frameworks. This feature facilitates the integration of policy improexvement steps with policy evaluation steps, and results in a single objective training process. Second, the normalizing constant of EBFlow is expressed in closed form, which enables the calculation of the soft value function without resorting to the approximation methods mentioned in Eqs. (6) and (7). Third, given that normalizing flow is a universal approximator for probability density functions, our policy's expressiveness is not constrained, and can model multi-modal action distributions.

In MEow, the policy is described as a state-conditioned EBFlow, with its pdf presented as follows:

$$
\begin{aligned}
\pi_\theta(\mathbf{a}_t|\mathbf{s}_t) &= \underbrace{p_\mathbf{z}\left(g_\theta(\mathbf{a}_t|\mathbf{s}_t)\right) \prod_{i\in\mathcal{S}_n} \left|\det\left(\mathbf{J}_{g_\theta^i}(\mathbf{a}_t^{i-1}|\mathbf{s}_t)\right)\right|}_{\text{Unnormalized Density}} \underbrace{\prod_{i\in\mathcal{S}_l} \left|\det(\mathbf{J}_{g_\theta^i}(\mathbf{s}_t))\right|}_{\text{Norm. Const.}} \\
&\triangleq \underbrace{\exp\left(\frac{1}{\alpha}Q_\theta(\mathbf{s}_t,\mathbf{a}_t)\right)}_{\text{Unnormalized Density}} \underbrace{\exp\left(-\frac{1}{\alpha}V_\theta(\mathbf{s}_t)\right)}_{\text{Norm. Const.}},
\end{aligned}
\tag{10}
$$

where the soft Q-function and the soft value function are selected as follows:

$$
Q_\theta(\mathbf{s}_t,\mathbf{a}_t) \triangleq \alpha\log p_\mathbf{z}\left(g_\theta(\mathbf{a}_t|\mathbf{s}_t)\right)\prod_{i\in\mathcal{S}_n}\left|\det\left(\mathbf{J}_{g_\theta^i}(\mathbf{a}_t^{i-1}|\mathbf{s}_t)\right)\right|, V_\theta(\mathbf{s}_t)\triangleq -\alpha\log\prod_{i\in\mathcal{S}_l}\left|\det(\mathbf{J}_{g_\theta^i}(\mathbf{s}_t))\right|.
\tag{11}
$$

Such a selection satisfies $V_\theta(\mathbf{s}_t) = \alpha\log\int\exp(Q_\theta(\mathbf{s}_t,\mathbf{a})/\alpha)d\mathbf{a}$ based on the property of EBFlow. In addition, both $Q_\theta$ and $V_\theta$ have a common output codomain $\mathbb{R}$, which enables them to learn to output arbitrary real values. These properties are validated in Proposition 3.1, with the proof provided in Appendix A.2. The training and inference processes of MEow are summarized as follows.

**Proposition 3.1.** *Eq. (11) satisfies the following statements: (1) Given that the Jacobian of $g_\theta$ is non-singular, $V_\theta(\mathbf{s}_t)\in\mathbb{R}$ and $Q_\theta(\mathbf{s}_t,\mathbf{a}_t)\in\mathbb{R}$, $\forall\mathbf{a}_t\in\mathcal{A},\forall\mathbf{s}_t\in\mathcal{S}$. (2) $V_\theta(\mathbf{s}_t) = \alpha\log\int\exp\left(Q_\theta(\mathbf{s}_t,\mathbf{a})/\alpha\right)d\mathbf{a}$.*

**Training.**   With $Q_\theta$ and $V_\theta$ defined in Eq. (11), the loss $\mathcal{L}(\theta)$ in Eq. (4) can be calculated without using Monte Carlo approximation of the soft value function target. Compared to the previous MaxEnt RL frameworks that rely on Monte Carlo estimation (i.e., Eqs. (6) and (7)), our framework offers the advantage of avoiding the errors induced by the approximation. In addition, MEow employs a unified policy rather than two separate roles (i.e., the actor and the critic), which eliminates the need for minimizing an additional policy improvement loss $\mathcal{L}(\phi)$ to bridge the gap between $\pi_\theta$ and $\pi_\phi$. This simplifies the training process of MaxEnt RL, and obviates the requirement of balancing between the two optimization loops.

**Inference.**   The sampling process of $\pi_\theta$ can be efficiently performed by deriving the inverse of $g_\theta$, as supported by several normalizing flow architectures [43–48]. In addition, unlike previous actor-critic frameworks susceptible to discrepancies between $\pi_\theta$ and $\pi_\phi$, the distribution established via $g_\theta^{-1}(\mathbf{z}|\mathbf{s}_t)$, where $\mathbf{z}\sim p_\mathbf{z}$, is consistently aligned with the pdf defined by $Q_\theta$. As a result, the actions taken by MEow can precisely reflect the learned soft Q-function.

## 3.2 Techniques for Improving the Training and Inference Processes of MEow

In this subsection, we introduce a number of training and inference techniques aimed at improving MEow while preserving its key features discussed in the previous subsection. For clarity, we refer to the MEow framework introduced in the last section as 'MEow (Vanilla)'.

**Learnable Reward Shifting (LRS).** Reward shifting [61–65] is a technique for shaping the reward function. This technique enhances the learning process by incorporating a shifting term in the reward function, which leads to a shifted optimal soft Q-function in MaxEnt RL. Inspired by this, this work proposes modeling a reward shifting function $b_\theta : \mathcal{S} \to \mathbb{R}$ with a neural network to enable the automatic learning of a reward shifting term. For notational simplicity, the parameters are denoted using $\theta$, and the details of the architecture are presented in Appendix A.5.1. The soft Q-function $Q_\theta^b$ augmented by $b_\theta$ is defined as follows:

$$Q_\theta^b(\mathbf{s}_t, \mathbf{a}_t) = Q_\theta(\mathbf{s}_t, \mathbf{a}_t) + b_\theta(\mathbf{s}_t). \quad (12)$$

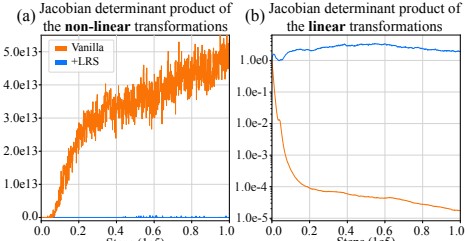

Figure 1: The Jacobian determinant products for (a) the non-linear and (b) the linear transformations, evaluated during training in the Hopper-v4 environment. Subfigure (b) is presented on a log scale for better visualization. This experiment adopt the affine coupling layers [47] as the nonlinear transformations.

The introduction of $Q_\theta^b$ results in a corresponding shifted soft value function $V_\theta^b(\mathbf{s}_t) \triangleq \alpha \log \int \exp(Q_\theta^b(\mathbf{s}_t, \mathbf{a})/\alpha) d\mathbf{a} = V_\theta(\mathbf{s}_t) + b_\theta(\mathbf{s}_t)$ (i.e., Proposition A.3 in Appendix A.2), which can be calculated without Monte Carlo estimation. Moreover, with the incorporation of $b_\theta$, the policy $\pi_\theta$ remains invariant since $\exp(\frac{1}{\alpha}(Q_\theta^b(\mathbf{s}_t, \mathbf{a}_t) - V_\theta^b(\mathbf{s}_t))) = \exp(\frac{1}{\alpha}((Q_\theta(\mathbf{s}_t, \mathbf{a}_t) + b_\theta(\mathbf{s}_t)) - (V_\theta(\mathbf{s}_t) + b_\theta(\mathbf{s}_t)))) = \exp(\frac{1}{\alpha}(Q_\theta(\mathbf{s}_t, \mathbf{a}_t) - V_\theta(\mathbf{s}_t)))$, which allows the use of $g_\theta^{-1}$ for efficiently sampling actions. As evidenced in Fig. 1, this method effectively addresses the issues of the significant growth and decay of Jacobian determinants of $g_\theta$ (discussed in Appendix A.3). In Section 4.4, we further demonstrate that the performance of MEow can be significantly improved through this technique.

**Shifting-Based Clipped Double Q-Learning (SCDQ).** As observed in [66], the overestimation of value functions often occurs in training. To address this issue, the authors in [66] propose clipped double Q-learning, which employs two separate Q-functions and uses the one with the smaller output to estimate the value function during training. This technique is also used in MaxEnt RL frameworks [9–13]. Inspired by this and our proposed learnable reward shifting, we further propose a shifting-based method that adopts two learnable reward shifting functions, $b_\theta^{(1)}$ and $b_\theta^{(2)}$, without duplicating the soft Q-function $Q_\theta$ and soft value function $V_\theta$ defined by $g_\theta$. The soft Q-functions $Q_\theta^{(1)}$ and $Q_\theta^{(2)}$ with corresponding learnable reward shifting functions $b_\theta^{(1)}$ and $b_\theta^{(2)}$ can be obtained using Eq. (12), while the soft value function $V_\theta^{\text{clip}}$ is written as the following formula:

$$V_\theta^{\text{clip}}(\mathbf{s}_t) = \min\left(V_\theta(\mathbf{s}_t) + b_\theta^{(1)}(\mathbf{s}_t), V_\theta(\mathbf{s}_t) + b_\theta^{(2)}(\mathbf{s}_t)\right) = V_\theta(\mathbf{s}_t) + \min\left(b_\theta^{(1)}(\mathbf{s}_t), b_\theta^{(2)}(\mathbf{s}_t)\right). \quad (13)$$

This design also prevents the production of two policies in MEow, as having two policies can complicate the inference procedure. In our ablation analysis presented in Section 4.4, we demonstrate that this technique can effectively improve the training process of MEow.

**Deterministic Policy for Inference.** As observed in [8], deterministic actors typically performed better as compared to its stochastic variant during the inference time. Such a problem can be formalized as finding an action $\mathbf{a}$ that maximizes $Q(\mathbf{s}_t, \mathbf{a})$ for a given $\mathbf{s}_t$. Since $\mathcal{A}$ is a continuous space, finding such a value would require extensive calculation. In the MEow framework, this value can be derived by making assumptions on the model architecture construction. Our key observation is that if the Jacobian determinants of the non-linearities (i.e., $g_\theta^i \in \mathcal{S}_n$) are constants with respect to its inputs, and that $\text{argmax}_\mathbf{z} \, p_\mathbf{z}(\mathbf{z})$ can be directly obtained, then the action $\mathbf{a}$ that maximizes $Q(\mathbf{s}_t, \mathbf{a})$ can be efficiently derived according to the following proposition.

**Proposition 3.2.** *Given that $|\det(\mathbf{J}_{g_\theta^i}(\mathbf{a}^{i-1}|\mathbf{s}_t))|$ is a constant with respect to $\mathbf{a}^{i-1}$, then $g_\theta^{-1}(\text{argmax}_\mathbf{z} \, p_\mathbf{z}(\mathbf{z})|\mathbf{s}_t) = \text{argmax}_\mathbf{a} \, Q_\theta(\mathbf{s}_t, \mathbf{a})$.*

The proof is provided in Appendix A.2. It is important to note that $g_\theta^i$ can still be a non-linear transformation, given that $|\det(\mathbf{J}_{g_\theta^i}(\mathbf{a}^{i-1}|\mathbf{s}_t))|$ is a constant. To construct such a model, a Gaussian prior with the additive coupling transformations [46] can be used as non-linearities. Under such a design, an action can be derived by calculating $g_\theta^{-1}(\mu|\mathbf{s}_t)$, where $\mu$ represents the mean of the

**Algorithm 1** Pseudo Code of the Training Process of MEow

**Input:** Learnable parameters $\theta$ and shadow parameters $\theta'$. Target smoothing factor $\tau$. Learning rate $\beta$.
Neural networks $g_\theta(\cdot\,|\,\cdot)$, $b_\theta^{(1)}(\cdot)$, and $b_\theta^{(2)}(\cdot)$. Temperature parameter $\alpha$. Discount factor $\gamma$.

1: **for** each training step **do**

2:     ▷ Extend the Replay Buffer.
3:     $\mathbf{a}_t = g_\theta^{-1}(\mathbf{z}|\mathbf{s}_t), \mathbf{z} \sim p_\mathbf{z}(\cdot)$.
4:     $\mathbf{s}_{t+1} \sim p_T(\cdot|\mathbf{s}_t, \mathbf{a}_t)$.
5:     $\mathcal{D} \leftarrow \mathcal{D} \cup \{(\mathbf{s}_t, \mathbf{a}_t, r_t, \mathbf{s}_{t+1})\}$.

6:     ▷ Update Policy.
7:     $(\mathbf{s}_t, \mathbf{a}_t, r_t, \mathbf{s}_{t+1}) \sim \mathcal{D}$.
8:     $Q_\theta(\mathbf{s}_t, \mathbf{a}_t) = \alpha \log(p_\mathbf{z}(g_\theta(\mathbf{a}_t|\mathbf{s}_t)) \prod_{i\in\mathcal{S}_n} |\det(\mathbf{J}_{g_\theta^i}(\mathbf{a}_t^{i-1}|\mathbf{s}_t))|)$.     ▷ Eq. (11)
9:     $V_{\theta'}(\mathbf{s}_{t+1}) = -\alpha \log \prod_{i\in\mathcal{S}_l} |\det(\mathbf{J}_{g_{\theta'}^i}(\mathbf{s}_{t+1}))|$.     ▷ Eq. (11)
10:    $Q_\theta^{(1)}(\mathbf{s}_t, \mathbf{a}_t) = Q_\theta(\mathbf{s}_t, \mathbf{a}_t) + b_\theta^{(1)}(\mathbf{s}_t)$ and $Q_\theta^{(2)}(\mathbf{s}_t, \mathbf{a}_t) = Q_\theta(\mathbf{s}_t, \mathbf{a}_t) + b_\theta^{(2)}(\mathbf{s}_t)$.     ▷ Eq. (12)
11:    $V_{\theta'}^{\text{clip}}(\mathbf{s}_{t+1}) = V_{\theta'}(\mathbf{s}_{t+1}) + \min\left(b_{\theta'}^{(1)}(\mathbf{s}_{t+1}), b_{\theta'}^{(2)}(\mathbf{s}_{t+1})\right)$.     ▷ Eq. (13)
12:    $\mathcal{L}(\theta) = \frac{1}{2}(Q_\theta^{(1)}(\mathbf{s}_t, \mathbf{a}_t) - (r_t + \gamma V_{\theta'}^{\text{clip}}(\mathbf{s}_{t+1})))^2 + \frac{1}{2}(Q_\theta^{(2)}(\mathbf{s}_t, \mathbf{a}_t) - (r_t + \gamma V_{\theta'}^{\text{clip}}(\mathbf{s}_{t+1})))^2$. ▷ Eq. (4)
13:    $\theta \leftarrow \theta + \beta\nabla_\theta\mathcal{L}(\theta)$.
14:    $\theta' \leftarrow (1-\tau)\theta' + \tau\theta$.

15: **end for**

Gaussian distribution. We elaborate on our model architecture design in Appendix A.5.1, and provide a performance comparison between MEow evaluated using a stochastic policy (i.e., $\mathbf{a}_t \sim \pi_\theta(\cdot\,|\,\mathbf{s}_t)$) and a deterministic policy (i.e., $\mathbf{a}_t = \mathrm{argmax}_\mathbf{a} Q_\theta(\mathbf{s}_t, \mathbf{a})$) in Section 4.4.

### 3.3 Algorithm Summary

We summarize the training process of MEow in Algorithm 1. The algorithm integrates the policy evaluation steps with the policy improvement steps, resulting in a single loss training process. This design differs from previous actor-critic frameworks, which typically perform two consecutive updates in each training step. In Algorithm 1, the learning rate is denoted as $\beta$. A set of shadow parameters $\theta'$ is maintained for calculating the delayed target values [67], and is updated according to the Polyak averaging [68] of $\theta$, i.e., $\theta' \leftarrow (1-\tau)\theta' + \tau\theta$, where $\tau$ is the target smoothing factor.

## 4 Experiments

In the following sections, we first present an intuitive example of MEow trained in a two-dimensional multi-goal environment [8] in Section 4.1. We then compare MEow's performance against several continuous-action RL baselines in five MuJoCo environments [32, 33] in Section 4.2. Next, in Section 4.3, we evaluate MEow's performance on a number of Omniverse Isaac Gym environments [34] simulated based on real-world robotic application scenarios. Lastly, in Section 4.4, we provide an ablation analysis to inspect the effectiveness of each proposed technique. Among all experiments, we maintain the same model architecture, while adjusting inputs and outputs according to the state space and action space for each environment. We construct $g_\theta$ using the additive coupling layers [46] with element-wise linear transformations, utilize a unit Gaussian as $p_\mathbf{z}$, and model the learnable adaptive reward shifting functions $b_\theta$ as multi-layer perceptrons (MLPs). For detailed descriptions of the experimental setups, please refer to Appendix A.5.

### 4.1 Evaluation on a Multi-Goal Environment

In this subsection, we present an example of MEow trained in a two-dimensional multi-goal environment [8]. The environment involves four goals, indicated by the red dots in Fig. 2 (a). The reward function is defined by the negative Euclidean distance from each state to the nearest goal, and the corresponding reward landscape is depicted using contours in Fig. 2 (a). The gradient map in Fig. 2 (a) represents the soft value function predicted by our model. The blue lines extending from the center represent the trajectories produced using our policy.

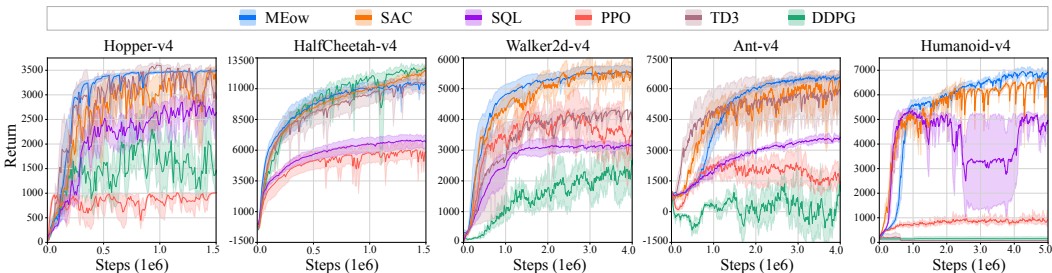

Figure 3: The results in terms of total returns versus the number of training steps evaluated on five MuJoCo environments. Each curve represents the mean performance, with shaded areas indicating the 95% confidence intervals, derived from five independent runs with different seeds.

As illustrated in Fig. 2 (a), our model's soft value function predicts higher values around the goals, suggesting successful learning of the goal positions through rewards. In addition, the trajectories demonstrate our agent's correct transitions towards the goals, which validates the effectiveness of our learned policy. To illustrate the potential impact of approximation errors that might emerge when employing previous soft value estimation methods, we compare three calculation methods for the soft value function: (I) Our approach (i.e., Eq. (11)): $V_\theta(\mathbf{s}_t)$, (II) SQL-like (i.e., Eq. (6)): $\alpha \log(\frac{1}{M} \sum_{i=1}^{M} \frac{\exp(Q_\theta(\mathbf{s}_t, \mathbf{a}^{(i)})/\alpha)}{\pi_\phi(\mathbf{a}^{(i)}|\mathbf{s}_t)})$,

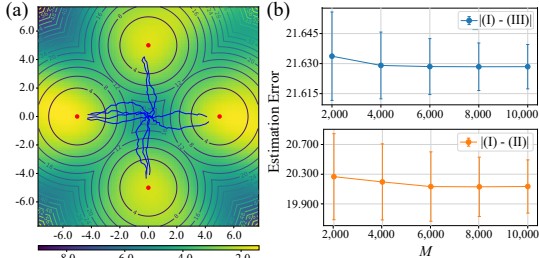

Figure 2: (a) The soft value function and the trajectories generated using our method on the multi-goal environment. (b) The estimation error evaluated at the initial state under different choices of $M$.

and (III) SAC-like (i.e., Eq. (7)): $\frac{1}{M} \sum_{i=1}^{M} (Q_\theta(\mathbf{s}_t, \mathbf{a}^{(i)}) - \alpha \log \pi_\phi(\mathbf{a}^{(i)}|\mathbf{s}_t))$, where $\{\mathbf{a}^{(i)}\}_{i=1}^{M}$ is sampled from $\pi_\phi$. The approximation errors of the soft value functions at the initial state are calculated using the Euclidean distances between (I) and (II), and between (I) and (III), for various values of $M$. As depicted in Fig. 2 (b), the blue line and the orange line decreases slowly with respect to $M$. These results suggest that Monte Carlo estimation converges slowly, making approximation methods such as Eqs. (6) and (7) challenging to achieve accurate predictions.

## 4.2 Performance Comparison on the MuJoCo Environments

In this experiment, we compare MEow with several commonly-used continuous control algorithms on five MuJoCo environments [32] from Gymnasium [33]. The baseline algorithms include SQL [8], SAC [9], deep deterministic policy gradient (DDPG) [69], twin delayed deep deterministic policy gradient (TD3) [66], and proximal policy optimization (PPO) [70]. The results for SAC, DDPG, TD3, and PPO were reproduced using Stable Baseline 3 (SB3) [71], utilizing SB3's refined hyperparameters. The results for SQL were reproduced using our own implementation, as SB3 does not support SQL and the official code is not reproducible. Our implementation adheres to SQL's original paper. Each method is trained independently under five different random seeds, and the evaluation curves for each environment are presented in the form of the means and the corresponding confidence intervals.

As depicted in Fig. 3, MEow performs comparably to SAC and outperforms the other baseline algorithms in most of the environments. Furthermore, in environments with larger action and state dimensionalities, such as 'Ant-v4' and 'Humanoid-v4', MEow offers performance improvements over SAC and exhibits fewer spikes in the evaluation curves. These results suggest that MEow is capable of performing high-dimensional tasks with stability. To further investigate the performance difference between MEow and SAC, we provide a thorough comparison between MEow, SAC [9], Flow-SAC [10, 11], and their variants in Appendix A.4.2. The results indicate that the training process involving policy evaluation and policy improvement steps may be inferior to our proposed training process with a single objective. In the next subsection, we provide a performance examination using the simulation environments from the Omniverse Isaac Gym [34].

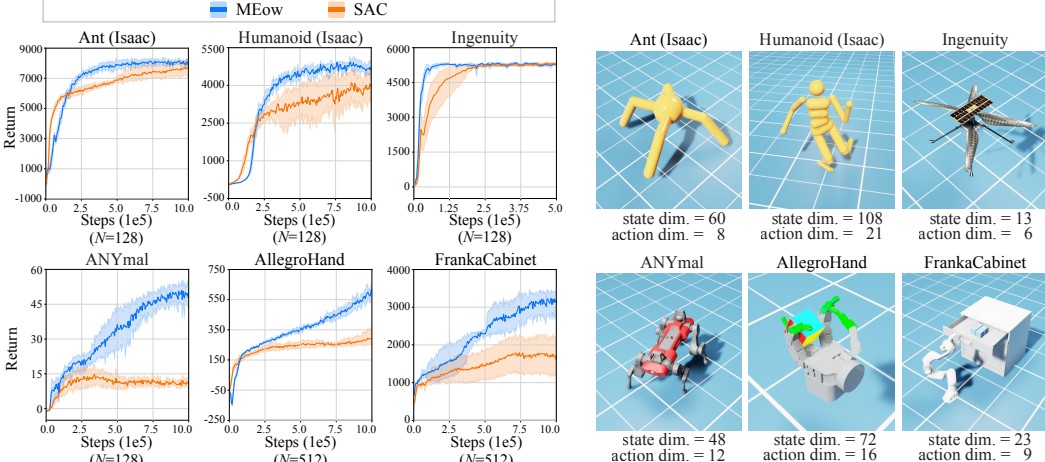

Figure 4: A comparison on six Isaac Gym environments. Each curve represents the mean performance of five runs, with shaded areas indicating the 95% confidence intervals. 'Steps' in the x-axis represents the number of training steps, each of which consists of $N$ parallelizable interactions with the environments.

Figure 5: A demonstration of the six Isaac Gym environments introduced in Section 4.3. The dimensionalities of the state and action for each environment are denoted below each subfigure.

## 4.3 Performance Comparison on the Omniverse Issac Gym Environments

In this subsection, we examine the performance of MEow on a variety of robotic tasks simulated by Omniverse Isaac Gym [34], a GPU-based physics simulation platform. In addition to 'Ant' and 'Humanoid', we employ four additional tasks: 'Ingenuity', 'ANYmal', 'AllegroHand', and 'Franka-Cabinet'. All of them are designed based on real-world robotic application scenarios. 'Ingenuity' and 'ANYmal' are locomotion environments inspired by NASA's Ingenuity helicopter and ANYbotics' industrial maintenance robots, respectively. On the other hand, 'AllegroHand' and 'FrankaCabinet' focus on executing specialized manipulative tasks with robotic hands and arms, respectively. A demonstration of these tasks is illustrated in Fig. 5.

In this experimental comparison, we adopt SAC as a baseline due to its excellent performance in the MuJoCo environments. The evaluation results are presented in Fig. 4. The results demonstrate that MEow exhibits superior performance on 'Ant (Isaac)' and 'Humanoid (Isaac)'. In addition, MEow consistently outperforms SAC across the four robotic environments (i.e., 'Ingenuity', 'ANYmal', 'AllegroHand', and 'FrankaCabinet'), indicating that our algorithm possesses the ability to perform challenging robotic tasks simulated based on real-world application scenarios.

## 4.4 Ablation Analysis

In this subsection, we provide an ablation analysis to examine the effectiveness of each technique introduced in Section 3.2.

**Training Techniques.** Fig. 6 compares the performance of three variants of MEow: 'MEow (Vanilla)', 'MEow (+LRS)', and 'MEow (+LRS & SCDQ)', across five MuJoCo environments. The results show that 'MEow (Vanilla)' consistently underperforms, with its total returns demonstrating negligible or no growth throughout the training period. In contrast, the variants incorporating translation functions demonstrate significant performance enhancements. This observation highlights the importance of including $b_\theta$ in the model design. In addition, the comparison between 'MEow (+LRS)' and 'MEow (+LRS & SCDQ)' suggests that our reformulated approach to clipped double Q-learning [66] improves the final performance by a noticeable margin.

**Inference Technique.** Fig. 7 compares the performance of two variants of MEow: 'MEow (Stochastic)' and 'MEow (Deterministic)'. The former samples action based on $\mathbf{a}_t \sim \pi_\theta(\cdot \,|\mathbf{s}_t)$ while the latter derive action according to $\mathbf{a}_t = \operatorname{argmax}_{\mathbf{a}} Q_\theta(\mathbf{s}_t, \mathbf{a}) = g_\theta^{-1}(\mu|\mathbf{s}_t)$. As shown in the figure, MEow with a deterministic policy outperforms its stochastic variant, suggesting that a deterministic policy may be more effective for MEow's inference.

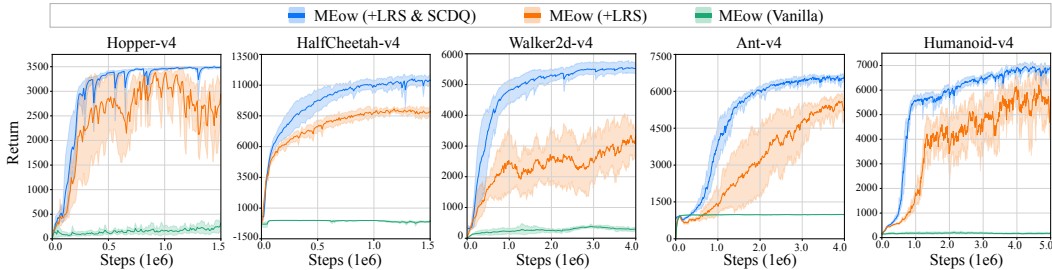

Figure 6: The performance comparison of MEow's variants (i.e., 'MEow (Vanilla)', 'MEow (+LRS)', and 'MEow (+LRS & SCDQ)') on five MuJoCo environments. Each curve represents the mean performance of five runs, with shaded areas indicating the 95% confidence intervals.

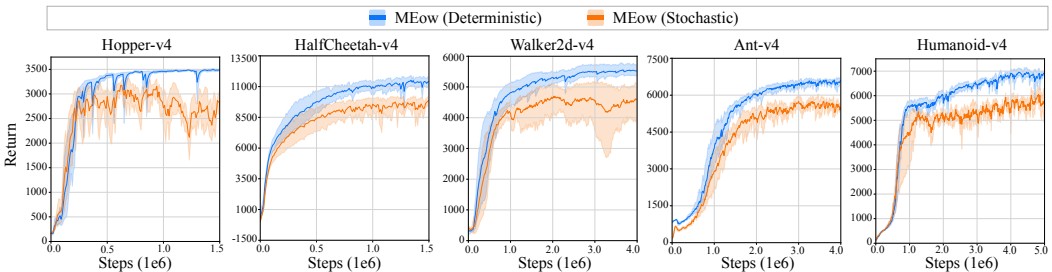

Figure 7: Performance comparison between MEow with a deterministic policy and MEow with a stochastic policy on five MuJoCo environments. Each curve represents the mean performance of five runs, with shaded areas indicating the 95% confidence intervals.

## 5  Conclusion

In this paper, we introduce MEow, a unified MaxEnt RL framework that facilitates exact soft value calculations without the need for Monte Carlo estimation. We demonstrate that MEow can be optimized using a single objective function, which streamlines the training process. To further enhance MEow's performance, we incorporate two techniques, learnable reward shifting and shifting-based clipped double Q-learning, into the design. We examine the effectiveness of MEow via experiments conducted in five MoJoCo environments and six robotic tasks simulated by Omniverse Isaac Gym. The results validate the superior performance of MEow compared to existing approaches.

### Limitations and Discussions

As discussed in Section 3.2, deterministic policies typically offer better performance compared to their stochastic counterparts. Although our implementation of MEow supports deterministic inference, this capability is based on the assumptions that the Jacobian determinants of the non-linear transformations are constants with respect to their inputs, and that $\text{argmax}_{\mathbf{z}} \, p_{\mathbf{z}}(\mathbf{z})$ can be efficiently derived. These assumptions may not hold for certain types of flow-based models. Therefore, exploring effective architectural choices for MEow represents a promising direction for further investigation.

On the other hand, the training speed of MEow is around $2.3\times$ slower than that of SAC, even though updates according to $\mathcal{L}(\phi)$ are bypassed in MEow. According to our experimental observations, the computational bottleneck of MEow may lie in the inference speed of the flow-based model during interactions with environments. While this speed is significantly faster than many iterative methods, such as MCMC or variational inference, it is still slower compared to the inference speed of Gaussian models. As a result, enhancing the inference speed of flow-based models represents a potential avenue for further improving the training efficiency of MEow.

Finally, our hyperparameter sensitivity analysis, as presented in A.4.5, indicates that our current approach requires different values of $\tau$ to achieve optimal performance. Since hyperparameter tuning often demands significant computational resources, establishing a more generalized parameter setting or developing an automatic tuning mechanism for $\tau$ presents an important direction for future exploration.

## Acknowledgement

The authors gratefully acknowledge the support from the National Science and Technology Council (NSTC) in Taiwan under grant numbers MOST 111-2223-E-002-011-MY3, NSTC 113-2221-E-002-212-MY3, and NSTC 113-2640-E-002-003. The authors would also like to express their appreciation for the computational resources from NVIDIA Corporation and NVIDIA AI Technology Center (NVAITC) used in this work. Furthermore, the authors extend their gratitude to the National Center for High-Performance Computing (NCHC) for providing the necessary computational and storage resources.

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

# A Appendix

In this Appendix, we begin with a discussion of the soft value estimation methods used in SQL and SAC in Section A.1. We then derive a number of theoretical properties of MEow in Section A.2. Next, we discuss the issue of numerical instability in Section A.3. Then, we present additional experimental results in Section A.4, and summarize the experimental setups in Section A.5. Finally, we elaborate on the potential impacts of this work in Section A.6.

## A.1 The Soft Value Estimation Methods in SAC and SQL

In this section, we elaborate on the soft value estimation methods mentioned in Section 2.2. We first show that $V_\theta(\mathbf{s}_t)$ approximated using Eq. (6) is greater than that approximated using Eq. (7) for any given state $\mathbf{s}_t$ in Proposition A.1 and Remark A.2. Then, we discuss their practical implementation.

**Proposition A.1.** *For any $\mathbf{s}_t \in \mathcal{S}$ and $\alpha \in \mathbb{R}_{>0}$, the following inequality holds:*

$$\alpha \log \mathbb{E}_{\mathbf{a} \sim \pi_\phi} \left[ \frac{\exp\left(Q_\theta(\mathbf{s}_t, \mathbf{a})/\alpha\right)}{\pi_\phi(\mathbf{a}|\mathbf{s}_t)} \right] \geq \mathbb{E}_{\mathbf{a} \sim \pi_\phi}[Q_\theta(\mathbf{s}_t, \mathbf{a}) - \alpha \log \pi_\phi(\mathbf{a}|\mathbf{s}_t)]. \tag{A1}$$

*Proof.*

$$\begin{aligned}
\alpha \log \mathbb{E}_{\mathbf{a} \sim \pi_\phi} \left[ \frac{\exp\left(Q_\theta(\mathbf{s}_t, \mathbf{a})/\alpha\right)}{\pi_\phi(\mathbf{a}|\mathbf{s}_t)} \right] &\overset{(i)}{\geq} \alpha \mathbb{E}_{\mathbf{a} \sim \pi_\phi} \left[ \log \left( \frac{\exp\left(Q_\theta(\mathbf{s}_t, \mathbf{a})/\alpha\right)}{\pi_\phi(\mathbf{a}|\mathbf{s}_t)} \right) \right] \\
&= \alpha \mathbb{E}_{\mathbf{a} \sim \pi_\phi} \left[ Q_\theta(\mathbf{s}_t, \mathbf{a})/\alpha - \log \pi_\phi(\mathbf{a}|\mathbf{s}_t) \right] \\
&= \mathbb{E}_{\mathbf{a} \sim \pi_\phi} \left[ Q_\theta(\mathbf{s}_t, \mathbf{a}) - \alpha \log \pi_\phi(\mathbf{a}|\mathbf{s}_t) \right],
\end{aligned}$$

where $(i)$ is due to Jensen's inequality. $\qquad\square$

*Remark* A.2. The inequality in Eq. (A1) preserves after applying the Monte Carlo estimation. Namely,

$$\alpha \log \left( \frac{1}{M} \sum_{i=1}^{M} \frac{\exp\left(Q_\theta(\mathbf{s}_t, \mathbf{a}^{(i)})/\alpha\right)}{\pi_\phi(\mathbf{a}^{(i)}|\mathbf{s}_t)} \right) \geq \frac{1}{M} \sum_{i=1}^{M} \left( Q_\theta(\mathbf{s}_t, \mathbf{a}^{(i)}) - \alpha \log \pi_\phi(\mathbf{a}^{(i)}|\mathbf{s}_t) \right), \tag{A2}$$

where $\{\mathbf{a}^{(i)}\}_{i=1}^{M}$ represents a set of samples drawn from $\pi_\phi$.

Unlike the estimation in Eq. (7), the estimation in Eq. (6) is guaranteed to converge to $V_\theta(\mathbf{s}_t)$ as $M \to \infty$. However, empirically, the estimation method in Eq. (7) is preferred and widely used in the contemporary MaxEnt framework. One potential reason could be the required number of samples needed for effective approximation. According to [8], $M = 32$ is an effective choice for Eq. (6), whereas $M = 1$ works well for Eq. (7), as adopted by many previous works. [9–12, 14].

## A.2 Theoretical Properties of MEow

In this section, we examine a number of key properties of the MEow framework. We begin by presenting a proposition to verify MEow's capability in modeling the soft Q-function and the soft value function. Then, we present a proposition to derive a deterministic policy in MEow. Finally, we offer a discussion of the impact of incorporating learnable reward shifting functions.

**Proposition 3.1** *Eq. (11) satisfies the following statements: (1) Given that the Jacobian of $g_\theta$ is non-singular, $V_\theta(\mathbf{s}_t) \in \mathbb{R}$ and $Q_\theta(\mathbf{s}_t, \mathbf{a}_t) \in \mathbb{R}$, $\forall \mathbf{a}_t \in \mathcal{A}, \forall \mathbf{s}_t \in \mathcal{S}$. (2) $V_\theta(\mathbf{s}_t) = \alpha \log \int \exp\left(Q_\theta(\mathbf{s}_t, \mathbf{a})/\alpha\right) d\mathbf{a}$.*

*Proof.* (1) Given that the Jacobian of $g_\theta$ is non-singular, $V_\theta(\mathbf{s}_t) \in \mathbb{R}$ and $Q_\theta(\mathbf{s}_t, \mathbf{a}_t) \in \mathbb{R}$, $\forall \mathbf{a}_t \in \mathcal{A}, \forall \mathbf{s}_t \in \mathcal{S}$.

If the Jacobian of $g_\theta$ is non-singular, then both $\prod_{i \in \mathcal{S}_l} |\det(\mathbf{J}_{g_\theta^i}(\mathbf{s}_t))| \in \mathbb{R}_{>0}$ and $\prod_{i \in \mathcal{S}_n} |\det(\mathbf{J}_{g_\theta^i}(\mathbf{a}_t^{i-1}|\mathbf{s}_t))| \in \mathbb{R}_{>0}$. This suggests that $\alpha \log \prod_{i \in \mathcal{S}_l} |\det(\mathbf{J}_{g_\theta^i}(\mathbf{s}_t))| \in \mathbb{R}$ and $\alpha \log p_{\mathbf{z}}\left(g_\theta(\mathbf{a}_t|\mathbf{s}_t)\right) \prod_{i \in \mathcal{S}_n} |\det(\mathbf{J}_{g_\theta^i}(\mathbf{a}_t^{i-1}|\mathbf{s}_t))| \in \mathbb{R}$. As a result, $V_\theta(\mathbf{s}_t) \in \mathbb{R}$ and $Q_\theta(\mathbf{s}_t, \mathbf{a}_t) \in \mathbb{R}$.

(2) $V_\theta(\mathbf{s}_t) = \alpha \log \int \exp\left(Q_\theta(\mathbf{s}_t, \mathbf{a})/\alpha\right) d\mathbf{a}$.

$$1 = \int \pi(\mathbf{a}|\mathbf{s}_t)d\mathbf{a} = \int p_\mathbf{z}\left(g_\theta(\mathbf{a}|\mathbf{s}_t)\right) \prod_{i \in \mathcal{S}_n} \left|\det\left(\mathbf{J}_{g_\theta^i}(\mathbf{a}^{i-1}|\mathbf{s}_t)\right)\right| \prod_{i \in \mathcal{S}_l} \left|\det(\mathbf{J}_{g_\theta^i}(\mathbf{s}_t))\right| d\mathbf{a}.$$

$$\Leftrightarrow \left(\prod_{i \in \mathcal{S}_l} \left|\det(\mathbf{J}_{g_\theta^i}(\mathbf{s}_t))\right|\right)^{-1} = \int p_\mathbf{z}\left(g_\theta(\mathbf{a}|\mathbf{s}_t)\right) \prod_{i \in \mathcal{S}_n} \left|\det\left(\mathbf{J}_{g_\theta^i}(\mathbf{a}^{i-1}|\mathbf{s}_t)\right)\right| d\mathbf{a}.$$

$$\Leftrightarrow \alpha \log \left(\prod_{i \in \mathcal{S}_l} \left|\det(\mathbf{J}_{g_\theta^i}(\mathbf{s}_t))\right|\right)^{-1} = \alpha \log \int p_\mathbf{z}\left(g_\theta(\mathbf{a}|\mathbf{s}_t)\right) \prod_{i \in \mathcal{S}_n} \left|\det\left(\mathbf{J}_{g_\theta^i}(\mathbf{a}^{i-1}|\mathbf{s}_t)\right)\right| d\mathbf{a}.$$

$$\Leftrightarrow -\alpha \log \prod_{i \in \mathcal{S}_l} \left|\det(\mathbf{J}_{g_\theta^i}(\mathbf{s}_t))\right| = \alpha \log \int p_\mathbf{z}\left(g_\theta(\mathbf{a}|\mathbf{s}_t)\right) \prod_{i \in \mathcal{S}_n} \left|\det\left(\mathbf{J}_{g_\theta^i}(\mathbf{a}^{i-1}|\mathbf{s}_t)\right)\right| d\mathbf{a}.$$

$$\Leftrightarrow V_\theta(\mathbf{s}_t) = \alpha \log \int \exp\left(Q_\theta(\mathbf{s}_t, \mathbf{a})/\alpha\right) d\mathbf{a}.$$

$\square$

In Section 3.2, we demonstrate that $\operatorname{argmax}_\mathbf{a} Q_\theta(\mathbf{s}_t, \mathbf{a})$ can be efficiently obtained through $g_\theta^{-1}(\operatorname{argmax}_\mathbf{z} p_\mathbf{z}(\mathbf{z})|\mathbf{s}_t)$. To provide theoretical support for this result, we include a proof for Proposition 3.2.

**Proposition 3.2** *Given that* $|\det(\mathbf{J}_{g_\theta^i}(\mathbf{a}^{i-1}|\mathbf{s}_t))|$ *is a constant with respect to* $\mathbf{a}^{i-1}$*, then* $g_\theta^{-1}(\operatorname{argmax}_\mathbf{z} p_\mathbf{z}(\mathbf{z})|\mathbf{s}_t) = \operatorname{argmax}_\mathbf{a} Q_\theta(\mathbf{s}_t, \mathbf{a})$.

*Proof.* Let $c \triangleq \prod_{i \in \mathcal{S}_n} |\det(\mathbf{J}_{g_\theta^i}(\mathbf{a}^{i-1}|\mathbf{s}_t))|$.

$$\operatorname*{argmax}_\mathbf{a} Q_\theta(\mathbf{s}_t, \mathbf{a}) = \operatorname*{argmax}_\mathbf{a} \alpha \log \left(p_\mathbf{z}\left(g_\theta(\mathbf{a}|\mathbf{s}_t)\right) \prod_{i \in \mathcal{S}_n} \left|\det\left(\mathbf{J}_{g_\theta^i}(\mathbf{a}^{i-1}|\mathbf{s}_t)\right)\right|\right)$$

$$= \operatorname*{argmax}_\mathbf{a} \alpha \log \left(p_\mathbf{z}\left(g_\theta(\mathbf{a}|\mathbf{s}_t)\right) c\right)$$

$$\overset{(i)}{=} \operatorname*{argmax}_\mathbf{a} p_\mathbf{z}\left(g_\theta(\mathbf{a}|\mathbf{s}_t)\right) c$$

$$= \operatorname*{argmax}_\mathbf{a} p_\mathbf{z}\left(g_\theta(\mathbf{a}|\mathbf{s}_t)\right)$$

$$\overset{(ii)}{=} \operatorname*{argmax}_\mathbf{a} p_\mathbf{z}\left(\mathbf{z}\right)$$

$$\overset{(iii)}{=} g_\theta^{-1}(\operatorname*{argmax}_\mathbf{z} p_\mathbf{z}(\mathbf{z})|\mathbf{s}_t),$$

where $(i)$ is because logarithm is strictly increasing, $(ii)$ is due to $\mathbf{z} = g_\theta(\mathbf{a}|\mathbf{s}_t)$, and $(iii)$ is due to $\mathbf{a} = g_\theta^{-1}(\mathbf{z}|\mathbf{s}_t)$. $\square$

In Section 3.2, we incorporate the learnable reward shifting function $b_\theta$ in $Q_\theta$ and $V_\theta$. This incorporation results in redefined soft Q-function $Q_\theta^b$ and soft value function $V_\theta^b$. In Proposition A.3, we verify that $V_\theta^b(\mathbf{s}_t) = \alpha \log \int \exp(Q_\theta^b(\mathbf{s}_t, \mathbf{a})/\alpha)d\mathbf{a} = V_\theta(\mathbf{s}_t) + b_\theta(\mathbf{s}_t)$.

**Proposition A.3.** *Given that $Q$ and $V$ satisfy $V(\mathbf{s}_t) \triangleq \alpha \log \int \exp\left(Q(\mathbf{s}_t, \mathbf{a})/\alpha\right) d\mathbf{a}$. The augmented functions, $Q^b(\mathbf{s}_t, \mathbf{a}_t) = Q(\mathbf{s}_t, \mathbf{a}_t) + b(\mathbf{s}_t)$ and $V^b(\mathbf{s}_t) \triangleq \alpha \log \int \exp(Q^b(\mathbf{s}_t, \mathbf{a})/\alpha)d\mathbf{a}$, where $b(\mathbf{s}_t)$ is the the reward shifting function, satisfy $V^b(\mathbf{s}_t) = V(\mathbf{s}_t) + b(\mathbf{s}_t)$.*

*Proof.*

$$V^b(\mathbf{s}_t) = \alpha \log \int \exp\left(Q^b(\mathbf{s}_t, \mathbf{a})/\alpha\right) d\mathbf{a}.$$

$$= \alpha \log \left( \int \exp\left((Q(\mathbf{s}_t, \mathbf{a}) + b(\mathbf{s}_t))/\alpha\right) d\mathbf{a} \right)$$

$$= \alpha \log \left( \exp(b(\mathbf{s}_t)/\alpha) \int \exp\left(Q(\mathbf{s}_t, \mathbf{a})/\alpha\right) d\mathbf{a} \right)$$

$$= \alpha \log \int \exp\left(Q(\mathbf{s}_t, \mathbf{a})/\alpha\right) d\mathbf{a} + \alpha \log \left(\exp(b(\mathbf{s}_t)/\alpha)\right)$$

$$= \alpha \log \int \exp\left(Q(\mathbf{s}_t, \mathbf{a})/\alpha\right) d\mathbf{a} + b(\mathbf{s}_t)$$

$$= V(\mathbf{s}_t) + b(\mathbf{s}_t).$$

$\square$

### A.3 The Issue of Numerical Instability

In this section, we provide the motivation for employing the learnable reward shifting function described in Section 3.2. We show that while $Q_\theta$ and $V_\theta$ defined in Eq. (11) have the theoretical capability to learn arbitrary real values (i.e., Proposition 3.1), they may experience numerical instability in practice. This instability arises due to the exponential growth of $\prod_{i \in \mathcal{S}_n} |\det(\mathbf{J}_{g_\theta^i}(\mathbf{a}_t^{i-1}|\mathbf{s}_t))|$ and the exponential decay of $\prod_{i \in \mathcal{S}_l} |\det(\mathbf{J}_{g_\theta^i}(\mathbf{s}_t))|$. We first examine the relationship between $V_\theta(\mathbf{s}_t)$ and $\prod_{i \in \mathcal{S}_l} |\det(\mathbf{J}_{g_\theta^i}(\mathbf{s}_t))|$ according to the following equations:

$$V_\theta(\mathbf{s}_t) = -\log \prod_{i \in \mathcal{S}_l} \left|\det\left(\mathbf{J}_{g_\theta^i}(\mathbf{s}_t)\right)\right| \quad \Leftrightarrow \quad \exp(-V_\theta(\mathbf{s}_t)) = \prod_{i \in \mathcal{S}_l} \left|\det\left(\mathbf{J}_{g_\theta^i}(\mathbf{s}_t)\right)\right|.$$

The above equation suggests that the value of $\prod_{i \in \mathcal{S}_l} |\det(\mathbf{J}_{g_\theta^i}(\mathbf{s}_t))|$ decreases exponentially with respect to $V_\theta(\mathbf{s}_t)$, which may lead to numerical instability during training. On the other hand, the relationship between $Q_\theta(\mathbf{s}_t, \mathbf{a}_t)$ and $\prod_{i \in \mathcal{S}_n} |\det(\mathbf{J}_{g_\theta^i}(\mathbf{a}_t^{i-1}|\mathbf{s}_t))|$ can be expressed according to the following equations:

$$Q_\theta(\mathbf{s}_t, \mathbf{a}_t) = \log p_\mathbf{z}\left(g_\theta(\mathbf{a}_t|\mathbf{s}_t)\right) + \log \prod_{i \in \mathcal{S}_n} \left|\det\left(\mathbf{J}_{g_\theta^i}(\mathbf{a}_t^{i-1}|\mathbf{s}_t)\right)\right|.$$

$$\Leftrightarrow \quad Q_\theta(\mathbf{s}_t, \mathbf{a}_t) - \log p_\mathbf{z}\left(g_\theta(\mathbf{a}_t|\mathbf{s}_t)\right) = \log \prod_{i \in \mathcal{S}_n} \left|\det\left(\mathbf{J}_{g_\theta^i}(\mathbf{a}_t^{i-1}|\mathbf{s}_t)\right)\right|.$$

$$\Leftrightarrow \quad \exp\left(Q_\theta(\mathbf{s}_t, \mathbf{a}_t) - \log p_\mathbf{z}\left(g_\theta(\mathbf{a}_t|\mathbf{s}_t)\right)\right) = \prod_{i \in \mathcal{S}_n} \left|\det\left(\mathbf{J}_{g_\theta^i}(\mathbf{a}_t^{i-1}|\mathbf{s}_t)\right)\right|.$$

The equations indicate that $\prod_{i \in \mathcal{S}_n} |\det(\mathbf{J}_{g_\theta^i}(\mathbf{a}_t^{i-1}|\mathbf{s}_t))|$ increases exponentially with respect to $Q_\theta(\mathbf{s}_t, \mathbf{a}_t) - \log p_\mathbf{z}\left(g_\theta(\mathbf{a}_t|\mathbf{s}_t)\right)$. Therefore, increasing $Q_\theta$ may also lead to an exponential growth of $\prod_{i \in \mathcal{S}_n} |\det(\mathbf{J}_{g_\theta^i}(\mathbf{a}_t^{i-1}|\mathbf{s}_t))|$.

LRS makes our model less susceptible to numerical calculation errors since the learnable reward shifting function $b_\theta$, unlike $Q_\theta$ and $V_\theta$, is not represented in logarithmic scale. Consider a case where FP32 precision is in use, 'MEow (Vanilla)' could fail to learn a target $V_{\theta^*}(\mathbf{s}_t) > 38, \forall \mathbf{s}_t$, since $\prod_{i \in \mathcal{S}_l} |\det(\mathbf{J}_{g_\theta^i}(\mathbf{s}_t))| = \exp(-V_{\theta^*}(\mathbf{s}_t)) < 2^{-126}$ cannot be represented using FP32 precision. Therefore, without shifting the reward function, the loss sometimes becomes undefined values, and can lead to ineffective training (e.g., the green lines in Fig. 6). The reward shifting term can be designed as a (state-conditioned) function or a (non-state-conditioned) value. It can also be learnable or non-learnable. All of these designs (i.e., $b_\theta(\mathbf{s}_t)$, $b(\mathbf{s}_t)$, $b_\theta$, and $b$) can be directly applied to MEow since none of them influences the action distribution. Based on our preliminary experiments, we identified that a learnable state-conditioned reward shifting delivers the best performance.

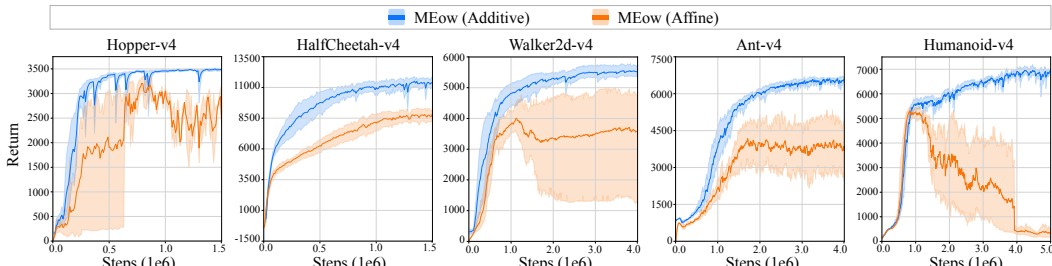

Figure A1: Performance comparison between MEow with additive coupling transformations in $g_\theta$ and MEow with affine coupling transformations in $g_\theta$ on five MuJoCo environments. Each curve represents the mean performance, with shaded areas indicating the 95% confidence intervals, derived from five independent runs with different seeds.

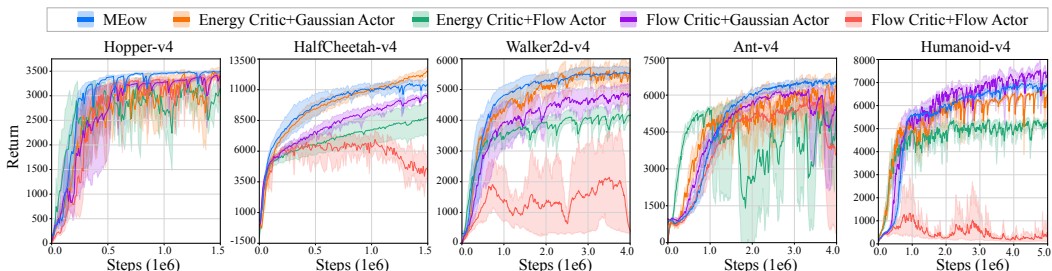

Figure A2: Performance comparison between 'MEow', 'Energy Critic+Gaussian Actor' (ECGA), 'Energy Critic+Flow Actor' (ECFA), 'Flow Critic+Gaussian Actor' (FCGA), and 'Flow Critic+Flow Actor' (FCFA) on five MuJoCo environments. Each curve represents the mean performance, with shaded areas indicating the 95% confidence intervals, derived from five independent runs with different seeds.

## A.4 Supplementary Experiments

In this section, we provide additional experimental results. In Section A.4.1, we offer a comparison between MEow with $g_\theta$ modeled using additive coupling layers and that using affine coupling layers. In Section A.4.2, we compare the performance of MEow with four distinct types of actor-critic frameworks formulated based on prior works [9–11]. In Section A.4.3, we provide an example illustrating the ability of flow-based models to represent multi-modal distributions as policies. In Section A.4.4, we present a performance comparison between SAC and its variant with LRS. Finally, in Section A.4.5, we provide a sensitivity examination for the target smoothing parameter.

### A.4.1 Comparison of Additive and Affine Transformations

In this section, we evaluate the performance of MEow with two commonly-adopted non-linear transformations, additive [46] and affine [47] coupling layers, for constructing $g_\theta$. The results are presented in Fig. A1. The results show that MEow with additive coupling layers achieves better performance than that with affine coupling layers. Based on this observation, we adopt additive coupling layers for constructing $g_\theta$ throughout the experiments in Section 4 of the main manuscript.

### A.4.2 Influences of Parameterization in MaxEnt RL Actor-Critic Frameworks

In this section, we compare the performance of MEow against four different actor-critic frameworks formulated based on prior works [9–11]. The first framework is the same as SAC [9], with the critic modeled as an energy-based model and the actor as a Gaussian. The second framework follows the approaches of [10, 11], where the critic is also an energy-based model, but the actor is a flow-based model. The third and fourth frameworks both utilize a flow-based model for the critic, with the actor modeled as a Gaussian and a flow-based model, respectively. These frameworks are denoted as: 'Energy Critic+Gaussian Actor' (ECGA), 'Energy Critic+Flow Actor' (ECFA), 'Flow Critic+Gaussian Actor' (FCGA), and 'Flow Critic+Flow Actor' (FCFA), respectively. Regarding the

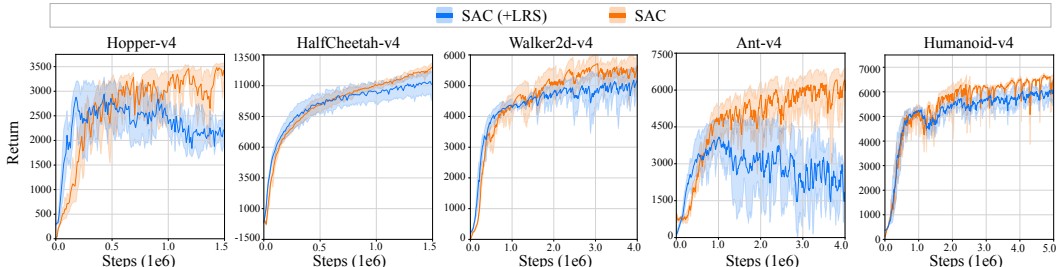

Figure A4: Performance comparison between 'SAC' and 'SAC (+LRS)'. Each curve represents the mean performance, with shaded areas indicating the 95% confidence intervals, derived from five independent runs with different seeds.

soft value calculation during training, the first and second frameworks adopt the value estimation method in SAC (i.e., Eq. (7)). For the third and the fourth frameworks, their training adopts the exact value calculation (i.e., Eq. (11)), which is the same as MEow. The results are presented in Fig. A2.

As depicted in Fig. A2, MEow exhibits superior performance and stability compared to other actor-critic frameworks in the 'Hopper-v4', 'Ant-v4', and 'Walker2d-v4' environments, and shows comparable performance with ECGA in most environments. In addition, the results that compare the frameworks with flow-based models as actors (i.e., ECFA and FCFA) to those with Gaussians as actors (i.e., ECGA and FCGA) suggest that Gaussians are more effective for modeling actors. This finding is similar to that in [10]. On the other hand, the comparisons between ECGA and FCGA, and between FCGA and FCFA, do not show a clear trend. These findings suggest that both flow-based and energy-based models can be suitable for modeling the soft Q-function. Furthermore, the comparison between FCFA and MEow reveals that the training process involving alternating policy evaluation and improvement steps may be inferior to our proposed training process with a single objective.

### A.4.3 Modeling Multi-Modal Distributions using Flow-based Models

In this section, we use a one-dimensional example to demonstrate that flow-based models are capable of learning multi-modal action distributions. We employ a state-conditioned neural spline flow (NSF) [50] as the model, and train it in a single-step environment with one-dimensional state and action spaces. Fig. A3 (a) illustrates the reward landscape with the state and action denoted on the x-axis and y-axis, respectively. Fig. A3 (b) illustrates the probability density function (pdf) predicted by the model. The result demonstrates the capability of flow-based models to effectively learn multi-modal distributions.

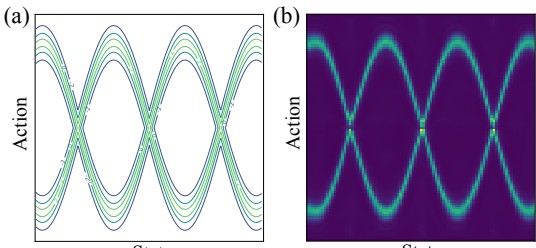

Figure A3: (a) The reward landscape of the one-step environment described in Section A.4.3. (b) The conditional pdf prediction using an NSF model.

### A.4.4 Applying Learnable Reward Shifting to SAC

In this section, we examine the performance of SAC with the proposed LRS technique. Since the original implementation of SAC involves the clipped double Q-Learning technique, SAC with LRS is equivalent to SAC with the shifting-based clipped double Q-Learning (SCDQ) technique discussed in Section 3.2 of the main manuscript. The performance of 'SAC' and 'SAC (+LRS)' is presented in Fig. A4. The results indicate that applying LRS does not improve SAC's performance. Therefore, for a fair evaluation, the original implementation of SAC is adopted in the comparison in Section 4 of the main manuscript.

### A.4.5 Sensitivity Examination for the Target Smoothing Parameter

In this section, we provide a performance comparison of SAC and MEow trained with different target smoothing parameter values (i.e., $\tau = 0.005, 0.003, 0.0005,$ and $0.0001$). The results shown in

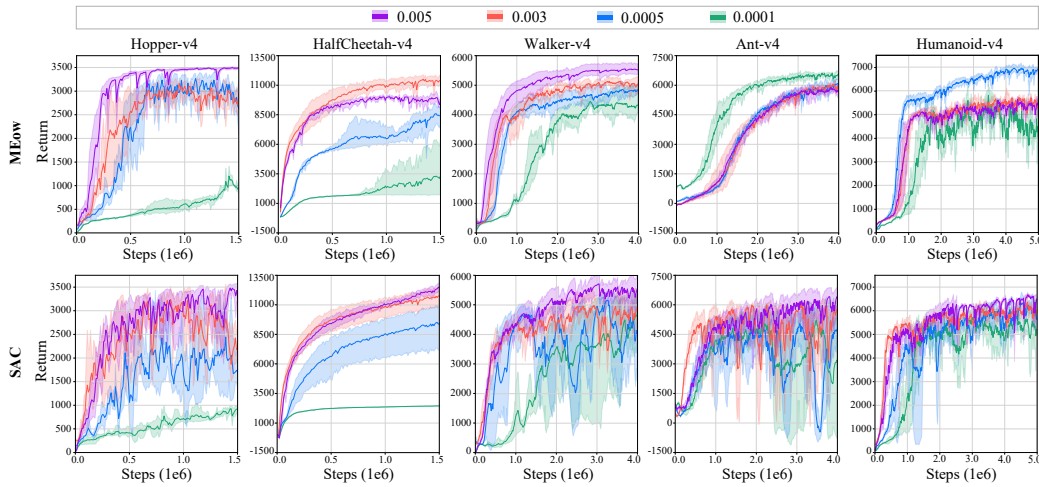

Figure A5: A performance comparison between MEow and SAC under different trained $\tau$ on five MuJoCo environments. Each curve represents the mean performance, with shaded areas indicating the 95% confidence intervals, derived from five independent runs with different seeds.

Fig. A5 indicate that SAC performs the best when $\tau = 0.005$, while MEow requires different $\tau$ values to achieve good performance across different tasks. Although both algorithms exhibit significant performance variations with different $\tau$ values, SAC demonstrates a more consistent trend in terms of the total returns among the tested values of $\tau$.

## A.5 Experimental Setups

In this section, we elaborate on the experimental configurations and provide the detailed hyperparameter setups for the experiments presented in Section 4 of the main manuscript. The code is implemented using PyTorch [72] and is available in the following repository: `https://github.com/ChienFeng-hub/meow`.

### A.5.1 Model Architecture

Among all experiments presented in Section 4 of the main manuscript, we maintain the same model architecture, while adjusting inputs and outputs according to the state space and action space for each environment. An illustration of this architecture is presented in Fig. A6. The model architecture comprises three main components: (I) normalizing flow, (II) hypernetwork, and (III) reward shifting function. For the first component, the transformation $g_\theta$ includes four additive coupling layers [46] followed by an element-wise linear layer. The prior distribution $p_{\mathbf{z}}$ is modeled as a unit Gaussian. For the second component, the hypernetwork involves two types of multi-layer perceptrons (MLPs), labeled as (a) and (b) in Fig. A6, which produce weights for the non-linear and linear transformations, respectively. Both MLPs employ swish activation functions [73] and have a hidden layer size of $64$. The MLPs labeled as (a) incorporate layer normalization [74] and a dropout layer [75] with a dropout rate of $0.1$. For the third component, the reward shifting functions (i.e., $b_\theta^{(1)}$ and $b_\theta^{(2)}$) are implemented using MLPs with swish activation and a hidden layer size of $256$. The parameters used in these components are collectively referred to as $\theta$, and are optimized using the same objective function $\mathcal{L}(\theta)$ defined in Eq. (4), with the soft Q-function and the soft value function replaced by $Q_\theta^b$ and $V_\theta^b$, respectively. Please note that, for the sake of notational simplicity and conciseness, the parameters of each network are all represented using $\theta$ instead of distinct symbols (e.g., $\theta_{\text{(II)-(a)}}$, $\theta_{\text{(II)-(b)}}$, $\theta_{\text{(III)-(1)}}$, and $\theta_{\text{(III)-(2)}}$).

### A.5.2 Experiments on the Multi-Goal Environment

The experiments in Section 4.1 are performed on a two-dimensional multi-goal environment [8]. This environment consists of four goals positioned at $[0, 5]$, $[0, -5]$, $[5, 0]$, and $[-5, 0]$, denoted as $\mathbf{g}_1$, $\mathbf{g}_2$, $\mathbf{g}_3$, and $\mathbf{g}_4$, respectively. The reward is the sum of two components, $r_1(\mathbf{s}_t)$ and $r_2(\mathbf{a}_t)$, which are

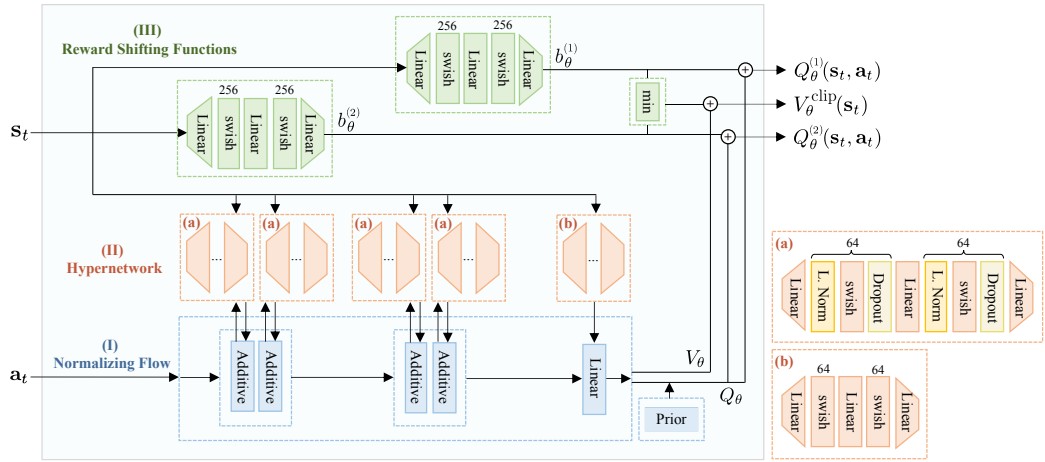

Figure A6: The architecture adopted in MEow. This architecture consists of three primary components: (I) normalizing flow, (II) hypernetwork, and (III) reward shifting function. The hypernetwork includes two distinct types of networks, labeled as (a) and (b), which are responsible for generating weights for the non-linear and linear transformations within the normalizing flow, respectively. Layer normalization is denoted as 'L. Norm' in (a).

<table>
<tr><td colspan="2">Table A1: Shared hyperparameters of MEow.</td></tr>
<tr><td>Parameter</td><td>Value</td></tr>
<tr><td>optimizer</td><td>Adam [78]</td></tr>
<tr><td>learning rate ($\beta$)</td><td>0.001</td></tr>
<tr><td>gradient clip value</td><td>30</td></tr>
<tr><td>discount ($\gamma$)</td><td>0.99</td></tr>
<tr><td>buffer size</td><td>$10^6$</td></tr>
</table>

| Table A1: Shared hyperparameters of MEow. | |
| --- | --- |
| Parameter | Value |
| optimizer | Adam [78] |
| learning rate ($\beta$) | 0.001 |
| gradient clip value | 30 |
| discount ($\gamma$) | 0.99 |
| buffer size | $10^6$ |

| Table A2: Shared hyperparameters of SAC. | |
| --- | --- |
| Parameter | Value |
| optimizer | Adam [78] |
| learning rate ($\beta$) | 0.0003 |
| gradient clip value | - |
| discount ($\gamma$) | 0.99 |
| buffer size | $10^6$ |

formulated as follow:

$$r_1(\mathbf{s}_t) = \max_i - \|\mathbf{s}_t - \mathbf{g}_i\| \text{ and } r_2(\mathbf{a}_t) = -30 \times \|\mathbf{a}_t\|. \tag{A3}$$

According to Eq. (A3), $r_1(\mathbf{s}_t)$ encourages policies to reach states near the goals. On the other hand, $r_2(\mathbf{a}_t)$ encourages policies to produce actions with small magnitudes.

In this experiment, we adopt a temperature parameter $\alpha = 2.5$, a target smoothing factor $\tau = 0.0005$, a learning rate $\beta = 0.001$, a discount factor $\gamma = 0.9$, and a total of $4,000$ training steps. The computation was carried out on NVIDIA TITAN V GPUs equipped with 12GB of memory. The training takes approximately four minutes.

### A.5.3 Experiments on the MuJoCo Environments

**Software and Hardware Setups.** For the experiments on the MuJoCo environments, our implementation is built on CleanRL [76], with the normalizing flow component adapted from [77]. The computation was carried out on NVIDIA V100 GPUs equipped with 16GB of memory. The training takes approximately 13 hours per 1 million steps, with each GPU capable of executing four training sessions simultaneously.

**Hyperparameter Setups.** The shared and the environment-specific hyperparameters of MEow are summarized in Tables A1 and A3, respectively. The hyperparamers for the baseline methods are directly borrowed from Stable Baseline 3 (SB3) [71].

Table A3: A list of environment-specific hyperparameters used in MEow.

|  | Environment | Target Smoothing Parameter ($\tau$) | Temperature Parameter ($\alpha$) |
|---|---|---|---|
| MuJoCo | Hopper-v4 | 0.005 | 0.25 |
|  | HalfCheetah-v4 | 0.003 | 0.25 |
|  | Walker2d-v4 | 0.005 | 0.1 |
|  | Ant-v4 | 0.0001 | 0.05 |
|  | Humanoid-v4 | 0.0005 | 0.125 |
| Omniverse Isaac Gym | Ant | 0.0005 | 0.075 |
|  | Humanoid | 0.00025 | 0.25 |
|  | Ingenuity | 0.0025 | 0.025 |
|  | ANYmal | 0.025 | 0.00075 |
|  | AllegroHand | 0.001 | 0.1 |
|  | FrankaCabinet | 0.075 | 0.1 |

Table A4: A list of environment-specific hyperparameters used in SAC.

|  | Environment | Target Smoothing Parameter ($\tau$) | Temperature Parameter ($\alpha$) |
|---|---|---|---|
| Omniverse Isaac Gym | Ant | 0.0025 | 0.4 |
|  | Humanoid | 0.0025 | 0.025 |
|  | Ingenuity | 0.0025 | 0.1 |
|  | ANYmal | 0.0025 | 0.01 |
|  | AllegroHand | 0.0025 | 0.1 |
|  | FrankaCabinet | 0.025 | 0.1 |

### A.5.4 Experiments on the Omniverse Isaac Gym Environments

**Software and Hardware Setups.** For the experiments performed on Omniverse Isaac Gym, the implementation is built on SKRL [79] due to its compatibility with Omniverse Issac Gym [34]. The computation was carried out on NVIDIA L40 GPUs equipped with 48GB of memory. The training takes approximately 22 hours per 1 million training steps, with each GPU capable of executing three training sessions simultaneously. For 'Ant', 'Humanoid', 'Ingenuity', and 'ANYmal', each training step consists of 128 parallelizable interactions with the environments. For 'AllegroHand' and 'FrankaCabinet', each training step consists of 512 parallelizable interactions with the environments.

**Hyperparameter Setups.** The shared and the environment-specific hyperparameters of MEow are summarized in Tables A1 and A3, respectively. Those of SAC are summarized in Tables A2 and A4, respectively. Both SAC and MEow were tuned using the same search space for $\tau$ and $\alpha$ to ensure a fair comparison. Specifically, a grid search was conducted with $\tau$ values ranging from 0.1 to 0.00025 and $\alpha$ values from 0.8 to 0.0005 for both algorithms. The setups with the highest average return were selected for each environment.

### A.6 Broader Impacts

This work represents a new research direction for MaxEnt RL. It discusses a unified method that can be trained using a single objective function and can avoid Monte Carlo estimation in the calculation of the soft value function, which addresses two issues in the existing MaxEnt RL methods. From a practical perspective, our experiments demonstrate that MEow can achieve superior performance compared to widely adopted representative baselines. In addition, the experimental results conducted in the Omniverse Isaac environments show that our framework can perform robotic tasks simulated based on real-world application scenarios. These results indicate the potential for deploying MEow in real robotic tasks. Given MEow's potential to be extended to perform challenging tasks, it is unlikely to have negative impacts on society.

