# OpenReview forum: "Maximum Entropy Reinforcement Learning via Energy-Based Normalizing Flow"
_NeurIPS.cc/2024/Conference — NeurIPS 2024 poster_

### Official Review · Reviewer_S3sW · 2024-07-10

**Soundness:** 3
**Presentation:** 3
**Contribution:** 3
**Rating:** 5
**Confidence:** 3

**Summary:**

This paper proposes a new framework for Maximum Entropy Reinforcement Learning using Energy-Based Normalizing Flows (EBFlow). The authors argue that this framework offers three main advantages: Firstly, the soft value function can be obtained without estimation. Second, the framework combines policy evaluation and policy improvement in one iteration. Finally, it does not use Monte Carlo methods and interacts with the environment effectively. To achieve this goal, Learnable Reward Shifting (LRS) technique which involves reward shifting for the value function using a reward shifting network and Shifting-Based Clipped Double Q-Learning (SCDQ) technique that employs two reward shifting networks for the value function and utilizes the minimum value function similar to double Q-learning have been proposed.

**Strengths:**

1. The paper is well-structured, and the overall content is easy to follow and intuitive. The detailed background explanations allow readers to progress through the main text with a solid understanding of the prior work.

2. The experimental domains are diverse, clearly demonstrating the strengths of the proposed algorithm in an easy-to-understand manner. The experiments are well-structured, and the detailed explanations make it easier for readers to follow the setup.

3. The proposed algorithm does not use Monte Carlo-based methods, allowing it to select the proper action for training during each step without waiting for the end of an episode. This enables the policy to interact with the environment more effectively

4. Existing SAC-based algorithms often rely on value function estimation, which can introduce errors. The proposed algorithm can obtain the soft value function without estimation, thus avoiding such errors.

**Weaknesses:**

1. Despite the fact that the proposed method is the first to apply EBFlow to reinforcement learning (RL), the novelty of the algorithm may appear weak, as there are several existing RL methods based on flow-based models. Besides applying EBFlow to the RL structure, were there are any additional considerations or advantages that were taken into account to enhance its performance or applicability in the context of RL? If such points exist, highlighting them would help to emphasize the contributions of the authors' proposed technique.

2. Introducing new frameworks generally necessitates providing sufficient mathematical proof to validate their efficacy, even for basic techniques or simple conjectures. It would be beneficial if the paper included mathematical proofs demonstrating that the proposed framework converges to the optimal policy. For example, the techniques presented in the paper, namely Learnable Reward Shifting (LRS) and Shifting-Based Clipped Double Q-Learning (SCDQ), have a significant impact on performance. However, the paper does not provide adequate mathematical justification for these techniques. Including such proofs would greatly enhance the confidence in the proposed method.

3. It would be beneficial for the paper to compare its methods with more recent state-of-the-art algorithms. The comparison with SAC, published in 2018, feels a bit outdated. Given that reinforcement learning is a rapidly evolving field, including comparisons with more recent methods would provide a clearer understanding of where this framework stands in the current landscape. Such comparisons would greatly enhance the paper by placing the proposed method in the context of the latest advancements.

**Questions:**

1. Please answer the questions in the weakness section

2. According to the appendix, the proposed method is said to be 2.3 times slower than SAC. Given this difference in speed, a time-wise comparison of the algorithm's performance would be very helpful to understand its efficiency relative to SAC.

3. In Figure 3, SAC and MEow show similar performance in the Mujoco domain, but there is a noticeable difference in performance in the Isaac Gym domain in Figure 4. Could you provide a further analysis on the reasons for the occurrence of this difference? It would provide valuable insight into the strengths and limitations of the proposed method.

Minor Issues\
Typo in line 149: "Jocobian" should be "Jacobian."\
Misreference in section 4.1: It should refer to Fig. 2, not Fig. 1.

**Limitations:**

They have handled it well.

---

> ### Author Rebuttal · Authors · 2024-08-06
>
> We thank the reviewer’s time and effort spent on the review, and would like to respond to the reviewer’s questions as follows.
>
> ---
>
> **Comments**
>
> **C1.** Despite the fact that the proposed method is the first to apply EBFlow to reinforcement learning (RL), the novelty of the algorithm may appear weak, as there are several existing RL methods based on flow-based models. ... If such points exist, highlighting them would help to emphasize the contributions of the authors' proposed technique.
>
> **Response:** As described in Lines 41-53 of Section 1 and Lines 166-174 of Section 3 of our paper, the proposed framework possesses the following two unique features. To the best of our knowledge, none of the existing MaxEnt RL frameworks, including those with flow-based models (e.g., [1-3]), possess the following characteristics:
>
> - Our framework enables the exact calculation of the soft value function, which improves the accuracy of the optimization process.
> - Our framework integrates the policy evaluation step and policy improvement step into a single training step, which streamlines the overall training process.
>
> Previous actor-critic frameworks without these features may suffer from inaccurate soft value estimation (i.e., Eqs. (6) and (7)) and estimation error when minimizing the policy improvement loss (i.e., Eq. (5)). We substantiate our claim through an extensive set of experiments conducted on the MuJoCo and Omniverse Isaac Gym benchmarks as depicted in Fig. 3 and 4 of the main manuscript.
>
> [1] Haarnoja et al. Latent Space Policies for Hierarchical Reinforcement Learning, ICML 2018.\
> [2] Mazoure et al. Leveraging exploration in off-policy algorithms via normalizing flows, CoRL 2019.\
> [3] Ward et al. Improving Exploration in Soft-Actor-Critic with Normalizing Flows Policies, ICML Workshop 2019.
>
> ---
>
> **C2.** Introducing new frameworks generally necessitates providing sufficient mathematical proof to validate their efficacy, even for basic techniques or simple conjectures. ... However, the paper does not provide adequate mathematical justification for these techniques. Including such proofs would greatly enhance the confidence in the proposed method.
>
> **Response:** We respectfully remind the reviewer that our method is mathematically supported by multiple theories.
>
> Our main methodology is theoretically supported by Proposition 3.1 in Lines 181-183, which verifies the validity of our selections of $Q_\theta$ and $V_\theta$. According to Theorem 3 in [1], which is described in Section 2.1, the policy of MEow converges to the optimal policy when the soft Bellman error in Eq. (4) is minimized.
>
> The theoretical justification for learnable reward shifting (LRS) is provided in Lines 212 to 219. Lines 214 and 215 show that the soft value function can be calculated exactly (Proposition A.3). In Lines 217 and 218, we mathematically demonstrate that the action distribution remains unchanged after applying LRS. The newly defined $Q_\theta^b$ and $V_\theta^b$ follow Theorem 3 in [1] and converge to the optimal policy when minimizing the soft Bellman error in Eq. (4).
>
> The theoretical justification for SCDQ is given in Lines 226 to 233. SCDQ allows for the implementation of clipped double Q-learning [2] without duplicating the policy in MEow. In Eq. (13) and Lines 228-230, we mathematically show how SCDQ prevents the introduction of additional soft value functions. The mechanisms by which clipped double Q-learning reduces training variances and overestimation are analyzed in TD3’s original paper [2].
>
> We would also like to emphasize that other techniques and claims in this work are theoretically supported with proofs. The deterministic inference technique is supported by Proposition A.4 in Lines 605-609. The theoretical analysis of the soft value estimation methods is offered in Proposition A.1 and Remark A.2 in Lines 565-568. These technical results support the validity of our method and provide motivation for formulating our approach.
>
> [1] Haarnoja et al. Reinforcement Learning with Deep Energy-Based Policies. ICML 2017.\
> [2] Fujimoto et al. Addressing Function Approximation Error in Actor-Critic Methods. ICML 2018.
>
> ---
>
> **C3.** It would be beneficial for the paper to compare its methods with more recent state-of-the-art algorithms. ... Such comparisons would greatly enhance the paper by placing the proposed method in the context of the latest advancements.
>
> **Response:** A performance comparison between MEow and two latest online RL frameworks, DIPO [1] published in 2023 and S2AC [2] published in 2024, is presented in Table 1 in the [global comment](https://openreview.net/forum?id=lhlIUxD5eE&noteId=Vapk6NMOyp). The results show that MEow outperforms DIPO and S2AC with noticeable margins in five MuJoCo environments.
>
> [1] Yang et al. Policy representation via diffusion probability model for reinforcement learning, 2023. \
> [2] Messaoud et al. S2AC: Energy-Based Reinforcement Learning with Stein Soft Actor-Critic, ICLR 2024.
>
> ---
>
> **Questions**
>
> **Q1.** According to the appendix, the proposed method is said to be 2.3 times slower than SAC. Given this difference in speed, a time-wise comparison of the algorithm's performance would be very helpful to understand its efficiency relative to SAC.
>
> **Response:** In response to the reviewer's request, we extended the training time of SAC and provided a timewise comparison between MEow and SAC evaluated on the Hopper-v4 environment in Fig. 3 of the attached PDF file in the [global comment](https://openreview.net/forum?id=lhlIUxD5eE&noteId=Vapk6NMOyp). The results show that MEow converges to a policy that achieves a better average return and exhibits greater stability.
>
> ---
>
> The response to **Q2** is provided in the [global comment](https://openreview.net/forum?id=lhlIUxD5eE&noteId=Vapk6NMOyp).

---

> > ### Comment · Reviewer_S3sW · 2024-08-12
> >
> > Thank you for providing the additional experiments and for addressing the questions with such a detailed rebuttal. Your response, along with the referenced paper, has more clarified the aspects I inquired about. I appreciate your efforts in improving the clarity of the work, and I will be increase my score accordingly.

---

> > > ### Author Response · Authors · 2024-08-12
> > >
> > > We sincerely appreciate the reviewer’s response and valuable feedback. Thank you again for your thoughtful review.

---

### Official Review · Reviewer_PzAH · 2024-07-10

**Soundness:** 3
**Presentation:** 2
**Contribution:** 3
**Rating:** 6
**Confidence:** 3

**Summary:**

This paper proposes to use the energy-based normalizing flow (EBFlow) to represent the policy and value functions in maximum entropy (MaxEnt) RL. Specifically, the paper builds the connection between EBFlow and MaxEnt RL by linking the conditional unnormalized density that depends on the input to the action value function Q and the constant that is independent of the input to the value function V. As a consequence, the Boltzmann policy defined by Q and V can be expressed exactly using a mapping in EBFlow. By updating the current Q estimate to approximate the action value of the Boltzmann policy, the algorithm would spontaneously update the Boltzmann policy. Hypothetically, the algorithm should converge to the optimal policy under entropy regularization. Using an existing implementation of EBFlow in the relevant literature, the paper proposed MEow based on this formulation. Experiments on four MuJoCo environments and six Issac Gym environments demonstrate the learning performance of MEow.

**Strengths:**

The strengths of this paper are as follows:
1. The connection between EBFlow and MaxEnt RL revealed in this paper is very interesting and, to my knowledge, novel. Given that MaxEnt is a popular approach in the RL literature, the issue addressed in this paper, learning the action value function and the Boltzmann policy, as well as the proposed approach, will be interesting to a broad set of audiences in the RL community.
2. The paper includes abundant comparative and ablation experiments. Notably, the ablation study covers the effect of the two components in the practical implementation, the difference between the deterministic and stochastic evaluation, etc.
3. The paper is well-written and mostly clear.

**Weaknesses:**

1. There is a potential concern about the empirical investigation: The proposed method is tuned over the target smoothing parameter, while the baselines are not tuned. Why is the target smoothing parameter tuned per environment? What would happen if the proposed method also used the same, fixed smoothing parameter?
2. This is a minor point. The organization of the paper can be improved. Specifically, the background section is too long, which takes up the space for more interesting discussions presented in the appendix. If space allows, the main text should include the network architecture in Figure A5, as it may be unfamiliar to a lot of readers. In addition, the limitation, especially the computational aspect of the proposed approach, should be discussed in the main paper.

Other suggestions that do not affect the evaluation:
* It would be great if there is at least some qualitative comparison between the proposed method and baselines using MaxEnt RL in the multi-goal environment.
* Typos between Lines 254 and 276: The reference to Figure 1 should be to Figure 2.

**Questions:**

Some questions that may affect the evaluation:
1. Are deterministic policies used for other algorithms in the evaluation in Figures 3 and 4?

**Limitations:**

The limitations of the work are discussed in the appendix, which should be moved to the main text. Another unmentioned potential limitation is the sensitivity to the target network smoothing parameter.

---

> ### Author Rebuttal · Authors · 2024-08-06
>
> We appreciate the reviewer’s valuable feedback and effort spent on the review and would like to respond to the reviewer’s questions as follows.
>
> ---
>
> **Comments**
>
> **C1.** **(1)** There is a potential concern about the empirical investigation: The proposed method is tuned over the target smoothing parameter, while the baselines are not tuned. **(2)** Why is the target smoothing parameter tuned per environment? What would happen if the proposed method also used the same, fixed smoothing parameter?
>
> **Response:**
> **(1)** Our baselines use the refined hyperparameters of Stable Baseline 3 (SB3) as mentioned in Line 285. Further modifications to the hyperparameters may not result in performance improvement. For a fair evaluation, we provide the results of SAC (the best-performing baseline) tuned with different target smoothing factors used in MEow (i.e., $\tau=0.005$, $0.003$, $0.0005$, and $0.0001$) in Fig. 1 of the attached PDF file in the [global comment](https://openreview.net/forum?id=lhlIUxD5eE&noteId=Vapk6NMOyp). The results show that SB3’s original hyperparameter ($\tau=0.005$) is the most effective.
>
> **(2)** We appreciate the reviewer's question regarding the target smoothing parameter $\tau$. According to Fig. 1 in the [global comment](https://openreview.net/forum?id=lhlIUxD5eE&noteId=Vapk6NMOyp), our current approach requires different values of $\tau$ to achieve good performance. However, we acknowledge the importance of determining a more generalized parameter setting. As part of our future work, we plan to expand our experimental framework to explore a broader range of $\tau$ values to identify a generally applicable $\tau$ across diverse environments.
>
> ---
>
> **C2.** This is a minor point. The organization of the paper can be improved. Specifically, the background section is too long, which takes up the space for more interesting discussions presented in the appendix. If space allows, the main text should include the network architecture in Figure A5, as it may be unfamiliar to a lot of readers. In addition, the limitation, especially the computational aspect of the proposed approach, should be discussed in the main paper.
>
> **Response:** We appreciate the reviewer's suggestion and concur that a more concise Section 2, coupled with the inclusion of discussions on MEow's model architecture and limitations in the main manuscript, will enhance the paper's quality. These modifications will be incorporated in the final revision.
>
> ---
>
> **C3.** Other suggestions.
> **(1)** It would be great if there is at least some qualitative comparison between the proposed method and baselines using MaxEnt RL in the multi-goal environment.
> **(2)** Typos between Lines 254 and 276: The reference to Figure 1 should be to Figure 2.
>
> **Response:**
> **(1)** We appreciate the reviewer's suggestion. A qualitative comparison among MEow, SAC, and SQL in the multi-goal environment is presented in Fig. 2 of the attached PDF file in the [global comment](https://openreview.net/forum?id=lhlIUxD5eE&noteId=Vapk6NMOyp).
> **(2)** We appreciate the reviewer’s attention to detail. The typo and figure reference will be corrected in the final revision.
>
> ---
>
> **C4.** **(1)** The limitations of the work are discussed in the appendix, which should be moved to the main text. **(2)** Another unmentioned potential limitation is the sensitivity to the target network smoothing parameter.
>
> **Response:**
> **(1)** We express our gratitude to the reviewer for the suggestions. A discussion of this work's limitations will be incorporated into the main manuscript in the final revision. **(2)** An examination of the sensitivity to the target network smoothing parameter is presented in Fig. 1 of the attachment to the [global comment](https://openreview.net/forum?id=lhlIUxD5eE&noteId=Vapk6NMOyp). This analysis will be included in the paper's final revision.
>
> ---
>
> **Questions**
>
> **Q1.** Are deterministic policies used for other algorithms in the evaluation in Figures 3 and 4?
>
> **Response:** During the evaluation phase, MEow, SAC, TD3, DDPG, and PPO employ deterministic policies. SQL, however, does not support deterministic inference and thus utilizes a stochastic policy for inference.

---

> > ### Comment · Reviewer_PzAH · 2024-08-09
> > **Follow-up questions on hyperparameter tuning**
> >
> > Thank you for the rebuttal and additional experiment results on SAC’s sensitivity on the target smoothing factor, $\tau$. I think the difference in the sensitivity to $\tau$ of SAC and MEow is very interesting and worth further investigation. In addition, I wonder what the hyperparameter tuning process is in the Issac gym environments. Are both SAC and MEow tuned (for both $\tau$ and $\alpha$)? If yes, how?

---

> > > ### Author Response · Authors · 2024-08-12
> > >
> > > We sincerely appreciate the reviewer’s response and valuable feedback. In the Isaac Gym experiments, both SAC and MEow were tuned using the same search space for $\tau$ and $\alpha$ to ensure a fair comparison. Specifically, a grid search was conducted with $\tau$ values ranging from 0.1 to 0.00025 and $\alpha$ values from 0.8 to 0.0005 for both algorithms. The setups with the highest average return were selected for each environment. We agree with the reviewer that the differences in the hyperparameters present an interesting direction for further investigation. Thank you again for your thoughtful review.

---

> > > > ### Comment · Reviewer_PzAH · 2024-08-12
> > > >
> > > > Thank you for the details. I encourage the author to present the full details about the Isaac Gym experiments, including SAC's environment-specific hyperparameters and the sensitivity analysis from the grid search. If these results are going to be included in the camera-ready version if accepted, I am willing to further raise my rating.

---

### Official Review · Reviewer_UfJ6 · 2024-07-11

**Soundness:** 3
**Presentation:** 3
**Contribution:** 3
**Rating:** 7
**Confidence:** 3

**Summary:**

The paper presents a new framework for Maximum-Entropy (MaxEnt) Reinforcement Learning (RL) using Energy-Based Normalizing Flows (EBFlow). Traditional MaxEnt RL methods, particularly for continuous action spaces, typically utilize actor-critic frameworks and alternate between policy evaluation and policy improvement steps. The proposed EBFlow framework integrates these steps into a single objective, eliminating the need for Monte Carlo approximation while enabling efficient modeling of multi-modal action distributions. The framework was experimentally validated on the MuJoCo benchmark and high-dimensional robotic tasks in Omniverse Isaac Gym, demonstrating superior performance compared to existing baseline methods.

**Strengths:**

- By combining policy evaluation and policy improvement into a single training process, the proposed method simplifies the optimization process and avoids potential optimization errors inherent in alternating updates.
- The use of normalizing flows facilitates efficient sampling and exact calculation of probability density functions, which addresses the inefficiencies associated with Monte Carlo methods in traditional MaxEnt RL approaches.
- Experimental results indicate that the EBFlow framework outperforms widely-adopted baselines on both the MuJoCo benchmark and complex robotic tasks, showcasing its effectiveness and robustness.

**Weaknesses:**

- There are some unreasonable aspects in the organization of the content. For example, Section 3.2 appears somewhat abrupt; it is suggested to move the content of A.4 into the main text. The discussion about inference in Section A.3 is also very important and should be included in the main text.
- The experimental section does not sufficiently highlight the advantages of the MEow method. The experiments in Section 4.1 on toy examples are not enough. Can comparisons be made from the perspectives of inference time, training time, representation capability, etc.?
- While the integration of policy evaluation and improvement steps simplifies the training process conceptually, the implementation details of the EBFlow framework may introduce additional complexity that needs to be managed.

**Questions:**

- GFlowNet points out that the MaxEnt RL paradigm has deficiencies in terms of diversity. Specifically, MaxEnt RL tends to achieve target states that are "further" from the initial state. Does MEow also inherit this defect of the MaxEnt RL paradigm?
- Energy-based normalizing flow is a generative model, and its original loss is to maximize log-likelihood. In Algorithm 1, this loss does not seem to be present. In my view, MEow uses the network structure of energy-based normalizing flow, but the loss still follows the MaxEnt RL paradigm. In this case, could the title of this paper be potentially misleading?
- Figure A5 shows that the MEow network uses a hypernetwork module. Why was this module introduced? Is this a common architecture for energy-based normalizing flow?
- What are the learning dynamics observed during the training process? Are there any notable differences in the convergence behavior compared to traditional actor-critic methods?
- How does the proposed framework scale with increasing complexity of the environment or the state-action space? Are there any specific limitations or challenges observed during experimentation?
- How robust is the EBFlow framework to different hyperparameters?

**Limitations:**

The author has already discussed the limitations ofthe method and its potential impacts in the paper, and provided possible solutions.

---

> ### Author Rebuttal · Authors · 2024-08-06
>
> We appreciate the reviewer’s valuable feedback and effort spent on the review, and would like to respond to the reviewer’s questions as follows.
>
> ---
>
> **Comments**
>
> **C1.** There are some unreasonable aspects in the organization of the content. For example, Section 3.2 appears somewhat abrupt; it is suggested to move the content of A.4 into the main text. The discussion about inference in Section A.3 is also very important and should be included in the main text.
>
> **Response:** We appreciate the reviewer's suggestion and would be glad to incorporate the contents of Sections A.3 and A.4 into the main manuscript to improve the paper's quality. We will implement this arrangement in the final revision.
>
> ---
>
> **C2.** The experimental section does not sufficiently highlight the advantages of the MEow method. The experiments in Section 4.1 on toy examples are not enough. Can comparisons be made from the perspectives of inference time, training time, representation capability, etc.?
>
> **Response:** We appreciate the reviewer’s suggestion and have expanded the experiments in Section 4.1 to include a comparison of inference and training times in the Humanoid environment (MuJoCo benchmark), as shown in the following table. The results demonstrate that performing Monte Carlo estimation and MCMC sampling at each step imposes a computational burden during both the training and inference phases.
>
> ||Training time (sec.)|Inference time (sec.)|
> |-|-|-|
> |MEow|0.022 $\pm$ 0.000|0.004 $\pm$ 0.000|
> |MC-based  ($M=10$)|0.110 $\pm$ 0.000|0.040 $\pm$ 0.004|
> |MC-based  ($M=100$)|1.064 $\pm$ 0.001|0.388 $\pm$ 0.004|
> |MC-based  ($M=200$)|3.181 $\pm$ 0.012|0.760 $\pm$ 0.004|
>
> **Table.** Training and inference time comparison. $M$ denotes the number of samples used for soft value estimation and the number of steps in the MCMC sampling process. The results are evaluated on NVIDIA V100 GPUs.
>
> Regarding the representation capability of our method, we have provided an analysis in Section A.5.4. The results indicate that our method possesses the ability to model multi-modal action distributions.
>
> ---
>
> **C3.** While the integration of policy evaluation and improvement steps simplifies the training process conceptually, the implementation details of the EBFlow framework may introduce additional complexity that needs to be managed.
>
> **Response:** The implementation of EBFlow can be simplified by integrating existing normalizing flow model libraries (e.g., [1,2]). Since log determinant calculation is already supported by many normalizing flow packages, the only required modification is the addition of an if-else condition during the forward pass to separate the energy function and the normalizing constant. We have made our code implementation available in an anonymous repository, with the corresponding link provided in Lines 699-700 of Section A.6.
>
> [1] Stimper et al. normflows: A PyTorch Package for Normalizing Flows, Journal of Open Source Software 2023.\
> [2] Durkan et al. nflows: normalizing flows in PyTorch. 2020.
>
> ---
>
> **Questions**
>
> **Q1.** GFlowNet points out that the MaxEnt RL paradigm has deficiencies in terms of diversity. Specifically, MaxEnt RL tends to achieve target states that are "further" from the initial state. Does MEow also inherit this defect of the MaxEnt RL paradigm?
>
> **Response:** We would like to highlight that the primary focus of this paper is on investigating the parameterization and its implications on MaxEnt RL. Comparing different RL frameworks (e.g., GFlowNet versus MaxEnt RL) falls outside the scope of this work. We acknowledge that further investigation into the topic of diversity within this context is an interesting direction for future research, and welcome any additional references the reviewer may provide to enrich this potential line of inquiry.
>
> ---
>
> **Q2.** Energy-based normalizing flow is a generative model, and its original loss is to maximize log-likelihood. ... but the loss still follows the MaxEnt RL paradigm. In this case, could the title of this paper be potentially misleading?
>
> **Response:** We believe that our paper title accurately reflects the key concept of the proposed methodology. As discussed in EBFlow’s original paper [1], a normalizing flow can be represented as an energy-based model and optimized using energy-based training objectives. Specifically, the authors in [1] employ score matching (which minimizes Fisher divergence) as an alternative to maximum likelihood training (which minimizes KL divergence) to optimize EBFlow. MEow can be viewed as an extension of EBFlow to the MaxEnt RL domain, where a policy is also represented as an energy-based model [2]. By synthesizing these concepts, we parameterize the policy using an EBFlow and demonstrate that the policy can be optimized by minimizing the soft Bellman errors.
>
> [1] Chao et al. Training Energy-Based Normalizing Flow with Score-Matching Objectives. NeurIPS 2023.\
> [2] Haarnoja et al. Reinforcement Learning with Deep Energy-Based Policies. ICML 2017.
>
> ---
>
> **Q3.** Figure A5 shows that the MEow network uses a hypernetwork module. Why was this module introduced? Is this a common architecture for energy-based normalizing flow?
>
> **Response:** Hypernetworks are commonly employed in conditional normalizing flow models for modeling the weights in the transformations (see [1] for more details). In our approach, the policy is modeled as a state-conditioned normalizing flow, with hypernetworks utilized to encode state information. As for implementation, we adopt the existing implementation of conditional normalizing flow [2], which incorporates a hypernetwork architecture.
>
> [1] Kobyzev et al. Normalizing Flows: An Introduction and Review of Current Methods. TPAMI 2019.\
> [2] Stimper et al. normflows: A PyTorch Package for Normalizing Flows, Journal of Open Source Software 2023.
>
> ---
>
> The responses to **Q4**-**Q6** are provided in the [global comment](https://openreview.net/forum?id=lhlIUxD5eE&noteId=Vapk6NMOyp).

---

> > ### Comment · Reviewer_UfJ6 · 2024-08-12
> >
> > Many thanks to the authors for thoroughly supplementing the key experiments and discussions. I have raised the score to 7.

---

> > > ### Author Response · Authors · 2024-08-12
> > >
> > > We sincerely appreciate the reviewer’s response and valuable feedback. Thank you again for your thoughtful review.

---

### Official Review · Reviewer_HRPY · 2024-07-12

**Soundness:** 4
**Presentation:** 4
**Contribution:** 3
**Rating:** 6
**Confidence:** 4

**Summary:**

The paper introduces a new MaxEnt RL framework called Meow based on Energy-based Normalizing Flow, which integrates the policy evaluation steps and the policy improvement steps and results in a single objective training process. Besides Meow enables the calculation of the soft value function used in the policy evaluation target without Monte Carlo approximation and supports the modeling of multi-modal action distributions.

**Strengths:**

1.	The proposed method is innovative and interesting, which utilizes the special property of Energy-based Normalizing Flow, and thus enables the consistency between the actor and critic and a single objective training process.
2.	The paper is overall well-written. The background and related works are introduced appropriately.
3.	The experimental results are comprehensive, involving the mujoco and Issac gym benchmarks.

**Weaknesses:**

1.	The calculation of the derterminant is usually time-consuming. So the reviewer wonders how the training time of Meow is compared with that of the classical off-policy RL methods like SAC.
2.	In the experiments, Meow is just compared with several classical RL methods for continuous control and lacks the comparison with advanced RL methods [1, 2] with other generative models like diffusion model and consistency model, which also supports the modeling of multi-modal action distribution.

[1] Yang L, Huang Z, Lei F, et al. Policy representation via diffusion probability model for reinforcement learning[J]. arXiv preprint arXiv:2305.13122, 2023.

[2] Yue Y, Kang B, Ma X, et al. Boosting offline reinforcement learning via data rebalancing[J]. arXiv preprint arXiv:2210.09241, 2022.

**Questions:**

1.	How is the training time of Meow compared with other methods?
2.	How is Meow compared with other advanced RL methods with the utilization of generative-model-based policy?
3.	Could the authors explain why Meow can learn a good reward shifting rather than a trivial shifting, like for any $s_t$, $b(s_t)=0$?

**Limitations:**

The authors have clearly presented the limitations in the paper.

---

> ### Author Rebuttal · Authors · 2024-08-06
>
> We appreciate the reviewer’s valuable feedback and effort spent on the review and would like to respond to the reviewer’s questions as follows.
>
> ---
>
> **Comments**
>
> **C1.** **(1)** The calculation of the determinant is usually time-consuming. **(2)** So the reviewer wonders how the training time of MEow is compared with that of the classical off-policy RL methods like SAC.
>
> **Response:**
> **(1)** Several existing normalizing flow architectures (e.g., [1] used in this work) support efficient calculation of Jacobian determinants, as discussed in Section 2.2 (Lines 138-140). In general, the Jacobian determinant of these architectures can be calculated with time complexity $O(D^2L)$, where $D$ represents the dimension of actions and $L$ is the number of layers in $g_\theta$ (defined in Line 132). This complexity is the same as forward passing a normalizing flow model.
>
> **(2)** A discussion regarding the training time of MEow and SAC is offered in Section A.7. Although the training time of our method is longer than that of SAC, SAC can not model multi-modal action distribution. For a fair comparison, we compared the training time of SAC-Flow (i.e., [2]) and MEow. We found that the training time of SAC-Flow is 1.13x slower than MEow due to the additional policy improvement updates (i.e., Eq. (5)).
>
> [1] Dinh et al. NICE: Non-linear Independent Components Estimation, ICLR Workshop 2015.\
> [2] Haarnoja et al. Latent Space Policies for Hierarchical Reinforcement Learning, ICML 2018.
>
> ---
>
> **C2.** In the experiments, MEow is just compared with several classical RL methods for continuous control and lacks the comparison with advanced RL methods [1, 2] with other generative models like diffusion model and consistency model, which also supports the modeling of multi-modal action distribution.
>
> **Response:** We appreciate the reviewer’s suggestion and have included a performance comparison between MEow and DIPO [1] in Table 1 in the [global comment](https://openreview.net/forum?id=lhlIUxD5eE&noteId=Vapk6NMOyp). The results show that MEow outperforms DIPO, even when DIPO uses 100 sequential forward passes during action sampling.
>
> To the best of our knowledge, consistency models have not been applied to online RL setups. In addition, combining consistency models with online MaxEnt RL may be challenging due to the intractability of the entropy calculation.
>
> On the other hand, reference [2] addresses offline RL tasks, which differ from the online RL setting discussed in this paper. The primary distinction is that in offline RL, agents cannot interact with the environment during training. Therefore, reference [2] represents a separate research direction and is distinct from the main focus of this paper.
>
> Finally, we would like to highlight that several online MaxEnt RL methods developed for modeling multi-modal policies are discussed in Section 2.2 and compared in Section 4.2. We use a number of representative MaxEnt RL methods, such as SQL, as our baselines in Fig. 3. We also compare our method with more advanced variants like SAC-Flow (i.e., [3]), denoted as ECFA in Fig. A3. Furthermore, Table 1 in the [global comment](https://openreview.net/forum?id=lhlIUxD5eE&noteId=Vapk6NMOyp) demonstrates that our method outperforms the latest MaxEnt online RL framework (i.e., S2AC [4]), where S2AC also supports modeling multi-modal action distributions.
>
> [1] Yang et al. Policy representation via diffusion probability model for reinforcement learning, 2023. \
> [2] Yue et al. Boosting offline reinforcement learning via data rebalancing, 2022.\
> [3] Haarnoja et al. Latent Space Policies for Hierarchical Reinforcement Learning, ICML 2018.\
> [4] Messaoud et al. S2AC: Energy-Based Reinforcement Learning with Stein Soft Actor-Critic, ICLR 2024.
>
> ---
>
> **Questions**
>
> **Q1.** How is the training time of MEow compared with other methods?
>
> **Response:** The training time comparison is discussed in the response of C1.
>
> ---
>
> **Q2.** How is MEow compared with other advanced RL methods with the utilization of generative-model-based policy?
>
> **Response:** Please refer to the responses to C2 and Table 1 in the [global comment](https://openreview.net/forum?id=lhlIUxD5eE&noteId=Vapk6NMOyp).
>
> ---
>
> **Q3.** Could the authors explain why MEow can learn a good reward shifting rather than a trivial shifting, like for any $s_t$, $b(s_t)=0$?
>
> **Response:** We appreciate the reviewer's question regarding MEow's ability to learn a good reward shifting rather than a trivial one. The inclusion of a learnable reward shifting term serves a crucial purpose: it prevents numerical errors in the Jacobian determinant calculations, as discussed in Section 3.2 and analyzed in Appendix A4. The efficacy of this approach is demonstrated in Fig. 1 of the main manuscript, where we observe the successful resolution of these numerical issues. Furthermore, Fig. 6 in the manuscript illustrates that this method contributes to improved overall performance. These results collectively indicate that MEow learns a meaningful and beneficial reward shifting, rather than a trivial one.
>
> In addition, according to our observation, $b_\theta$ does not learn to be a constant. Its value changes according to different visited states. To verify this, we sample a trajectory of states and plot the corresponding values of $b_\theta$ for the Hopper-v4 environment during the inference phase in Fig. 5 of the attached PDF file in the [global comment](https://openreview.net/forum?id=lhlIUxD5eE&noteId=Vapk6NMOyp).

---

> > ### Comment · Reviewer_HRPY · 2024-08-09
> >
> > Thanks for your responses and the efforts on the additional experiments. However, as shown in Table 2 of "Boosting offline reinforcement learning via data rebalancing", the experiments in online RL setting are also considered. In that case, the reviewer thinks this paper cannot be viewed as separate research work from yours. However, the contribution and novelty of this paper should be acknowledged. Hence, I am sticking to my score.

---

> > > ### Author Response · Authors · 2024-08-12
> > >
> > > We sincerely appreciate the valuable feedback and recognition of our work's contribution and novelty. Thank you once again for your thorough review.

---

### Official Review · Reviewer_he3t · 2024-07-13

**Soundness:** 3
**Presentation:** 3
**Contribution:** 4
**Rating:** 7
**Confidence:** 3

**Summary:**

The paper presents a new method for MaxEnt RL based on the recently proposed energy-based normalizing flows (EBFlow). The adoption of EBFlow allows us to overcome two major issues with training MaxEnt RL algorithms: (I) sampling from an energy-based model (of the policy) and (ii) approximating the soft value function (with MC methods).

**Strengths:**

* **Contribution**: I think the contribution of this work is significant and novel. The issues discussed with MaxEnt RL are important. The idea proposed is novel, it naturally emerges from the EBFlow framework and it's seamlessly applied to the MaxEnt RL domain. The approach solves the problems mentioned and it demonstrates superior performance compared to other standard approaches.
* **Presentation**: the presentation is clear and well-organised, and the method is described in detail.

**Weaknesses:**

* **New and old challenges**: while the approach resolves some issues with the current MaxEnt RL methods, it still suffers from typical deep RL issues (e.g. overestimation, for which the authors proposed SCDQ) and it introduces additional challenges (e.g. exploding Jacobian determinants, for which the authors introduce LRS). Normalizing flows also introduce additional constraints on the network's architecture, a problem which I believe is strongly mitigated by the introduced LRS (?)
* **Required clarification on LRS**: see Questions

**Questions:**

* Typo at line 149, "Jocobian"
* I am not sure if the role of the Learnable Reward Shifting (LRS) is clear to me. The authors say that they were inspired by the reward-shifting literature. However, I don't see the LRS term working the same way as classical reward shaping, as a state-dependent non-linear function ($b_\theta$), without any particular constraints (I don't see any stated) wouldn't preserve the properties of the original functions. I think the role of the learnable reward shifting is to have some form of "baseline" in the Q and V functions that simplifies the estimation process (since it's using a more flexible learning architecture), and so it avoids the explosion of the Jacobian. Could the authors provide clarification on this?
* I would be interested in seeing a comparison between the learned reward shifting term and the value function (V) in a plot (it would suffice to see this in the Appendix)

**Limitations:**

Limitations are addressed in Appendix. I think the authors should mention this in the main text.

---

> ### Author Rebuttal · Authors · 2024-08-06
>
> We appreciate the reviewer’s valuable feedback and effort spent on the review and would like to respond to the reviewer’s questions as follows.
>
> ---
>
> **Questions**
>
> **Q1.** Typo at line 149, "Jocobian".
>
> **Response:** Thank you for pointing this out. We will correct the typo in the final revision.
>
> ---
>
> **Q2.** I am not sure if the role of the Learnable Reward Shifting (LRS) is clear to me. The authors say that they were inspired by the reward-shifting literature. However, I don't see the LRS term working the same way as classical reward shaping, as a state-dependent non-linear function ($b_\theta$), without any particular constraints (I don't see any stated) wouldn't preserve the properties of the original functions. I think the role of the learnable reward shifting is to have some form of "baseline" in the Q and V functions that simplifies the estimation process (since it's using a more flexible learning architecture), and so it avoids the explosion of the Jacobian. Could the authors provide clarification on this?
>
> **Response:** Based on our formulation, the original $Q_\theta$ and $V_\theta$ functions are the residuals between their respective targets and $b_\theta$. We agree with the reviewer that this property of $b_\theta$ helps $Q_\theta$ and $V_\theta$ in predicting their targets during the optimization process.
>
> Another important aspect of LRS is that it reduces the magnitude of the soft value function $V_\theta$, which makes our model less susceptible to numerical calculation errors. An analysis of this issue is provided in Section A4. This problem arises due to limited numerical precision. Consider a case where FP32 precision is in use, `MEow (Vanilla)’ could fail to learn a target $V_\{\theta^{\*}\} (s_t)>38, \forall s_t$, since $\exp(-V_\{\theta^{\*}\}(s_t))=\Pi_\{i \in S_l\} \| \det (J_\{g_\{\theta^{\*}\}^i\}(s_t)) |<2^{-126}$ cannot be represented using FP32 precision. Therefore, without shifting the reward function, the loss sometimes becomes undefined values, and can lead to ineffective training (e.g., the green lines in Fig. 6). The reward shifting term can be designed as a (state-conditioned) function or a (non-state-conditioned) value. It can also be learnable or non-learnable. All of these designs (i.e.,  $b_\{\theta\} (s_t)$,  $b\(s_t\)$,  $b_\{\theta\}$,  and $b$) can be directly applied to MEow since none of them influences the action distribution. Based on our preliminary experiments, we identified that a learnable state-conditioned reward shifting worked the best and proposed this technique.
>
> ---
>
> **Q3.** I would be interested in seeing a comparison between the learned reward shifting term and the value function (V) in a plot.
>
> **Response:** We appreciate the reviewer's suggestion. The plot is offered in Fig. 4 of the attached PDF file in the [global comment](https://openreview.net/forum?id=lhlIUxD5eE&noteId=Vapk6NMOyp).

---

> > ### Comment · Reviewer_he3t · 2024-08-12
> >
> > I am satisfied with the authors' response, which clarifies my main concerns with the work.
> >
> > I think that a more appropriate description of how LRS is useful should be included in the paper, to improve clarity about the authors' contribution. A description similar to the one provided here would suffice.
> >
> > The additional results presented also seem to reflect my original understanding and the authors' response about how LRS works and its useful in their method.
> >
> > I increased my score to 7 and will recommend acceptance of the work.

---

> > > ### Author Response · Authors · 2024-08-12
> > >
> > > We sincerely appreciate the reviewer’s response and valuable feedback. We would be glad to incorporate the additional explanation of LRS into the manuscript in the final revision. Thank you again for your thoughtful review.

---

### Author Rebuttal · Authors · 2024-08-06

This global comment includes additional experimental results and extended discussions addressing the questions raised by reviewers UfJ6 and S3sW.

---

### **Additional Results**

The attached PDF file contains five figures, denoted as **Figs. 1-5**, which encompass the following content:

- **Fig. 1**: A performance comparison of MEow and SAC trained with different target smoothing values ($\tau$).
- **Fig. 2**: The soft value functions and the trajectories generated using MEow, SAC, and SQL on the multi-goal environment.
- **Fig. 3**: The return versus training time comparison of MEow and SAC.
- **Fig. 4**: $V_\theta$ and $b_\theta$ of MEow evaluated on the multi-goal environment.
- **Fig. 5**: The learnable reward shifting values evaluated along a trajectory of states in the Hopper-v4 environment.

The following table compares the performance of MEow and a number of recent online RL methods:

||Hopper|HalfCheetah|Walker|Ant|Humanoid|
|-|-|-|-|-|-|
|MEow|**3332.99** $\pm$ 521.63|**10981.47**$\pm$1812.97|**5526.66** $\pm$ 276.99|**6586.33**$\pm$188.73|**6923.22**$\pm$125.93|
|DIPO [1] ($K$=100)|3123.14$\pm$ 636.23|10472.31$\pm$654.96|4409.18$\pm$469.06|5622.30$\pm$487.09|4878.41$\pm$822.03|
|DIPO [1] ($K$=50)|3214.83$\pm$491.15|9198.20$\pm$1738.25|4199.34$\pm$ 1062.31|4877.41 $\pm$ 1010.35|4513.39$\pm$ 1075.94|
|DIPO [1] ($K$=20)|2511.63 $\pm$ 837.03|9343.69 $\pm$ 986.82|4467.20 $\pm$ 368.13|5288.77 $\pm$ 970.35|4294.79 $\pm$ 1583.48|
|S2AC [2]|<3100|<10000|<3500|<3000|<3500|

**Table 1.** Comparison of the average return between MEow, DIPO [1], and S2AC [2]. The results for DIPO [1] are obtained directly from Table 1 of their original paper. $K$ represents the number of diffusion steps used in the sampling process of [1]. The results for S2AC [2] are derived from Fig. 8 of their original paper.

[1] Yang et al. Policy representation via diffusion probability model for reinforcement learning, 2023.\
[2] Messaoud et al. S2AC: Energy-Based Reinforcement Learning with Stein Soft Actor-Critic, ICLR 2024.

---

### **Reviewer UfJ6 (Cont'd)**

**Q4.** What are the learning dynamics observed during the training process? Are there any notable differences in the convergence behavior compared to traditional actor-critic methods?

**Response:** In Section A.5.3 of the Appendix, we compare MEow with four different types of actor-critic frameworks. Our observations indicate that MEow converges to policies that achieve higher returns in most of the environments. In particular, the comparison between MEow and FCFA (i.e., an actor-critic framework modeled using two normalizing flows) suggests that the integration of policy improvement and evaluation steps enhances training stability and overall performance. This finding substantiates the primary claim of our paper.

---

**Q5.** How does the proposed framework scale with increasing complexity of the environment or the state-action space? Are there any specific limitations or challenges observed during experimentation?

**Response:** Both the loss function calculation and the inference process of MEow are scalable with respect to the state-action space. This scalability is attributable to EBFlow's compatibility with existing normalizing flow architectures that support efficient inverse transformation and Jacobian determinant calculation for high-dimensional inputs. These architectures can be utilized in MEow to facilitate efficient training and inference processes with a large action space. For environments with a large state space, states can be encoded using hypernetworks (e.g., MLP in our implementation) to reduce their dimensionality, thus ensuring scalability. Our experiments in high-dimensional environments, such as Humanoid (state dim.: 108; action dim.: 21) and AllegroHand (state dim.: 72; action dim.: 16), support this claim.

---

**Q6.** How robust is the EBFlow framework to different hyperparameters?

**Response:** We appreciate the reviewer’s suggestion. In response, we conducted a performance comparison under different values of $\tau$ (i.e., smoothing target factor in Section 3.3) on five MuJoCo environments. The results of this analysis are presented in Fig. 1 in the attachment of this global comment.

---

### **Reviewer S3sW (Cont'd)**

**Q2.** In Figure 3, SAC and MEow show similar performance in the Mujoco domain, but there is a noticeable difference in performance in the Isaac Gym domain in Figure 4. Could you provide a further analysis on the reasons for the occurrence of this difference? It would provide valuable insight into the strengths and limitations of the proposed method.

**Response:** The difference in performance between SAC and MEow in the MuJoCo and Isaac Gym environments can be attributed to the varying complexity and nature of tasks in these environments.

In the MuJoCo domain, the five tasks exhibit relative homogeneity, primarily focusing on locomotion. These tasks involve straightforward action outputs, typically torque. On the other hand, the Isaac Gym domain presents a more diverse and complex set of tasks. For instance, the ANYmal robot, unlike HalfCheetah, must navigate in multiple directions and follow specific target velocities. Its action space is different as well, involving outputs related to joint position targets rather than torque. Dexterous manipulations, such as those required in AllegroHand tasks, are widely recognized as challenging in the realm of RL. These tasks entail intricate interactions and contacts between the hand and the objects it manipulates. While SAC performs comparably to MEow in the relatively simple MuJoCo tasks, MEow demonstrates greater robustness and generalizability in the more diverse and complex tasks of the Isaac Gym domain. This highlights the strengths of MEow in handling a wider range of challenging scenarios.

---

### Decision · Program_Chairs · 2024-09-25

**Decision:**

Accept (poster)

**Comment:**

The paper proposes a new MaxEnt Reinforcement Learning (RL) framework using Energy-Based Normalizing Flows (EBFlow). Traditional MaxEnt RL methods typically use actor-critic frameworks and alternate between policy evaluation and policy improvement steps. The proposed framework integrates these steps into a single objective, eliminating the need for Monte Carlo approximation while enabling efficient modeling of multi-modal action distributions. Experiments on the MuJoCo benchmark and Omniverse Isaac Gym environments show superior performance compared to existing MaxEnt RL methods.

**Strengths**
- The idea of adopting EBFlow for MaxEnt RL is novel.
- The presentation is clear and well-organized, and the method is described in detail.
- The experimental results are convincing.

**Weaknesses**
- The calculation of the determinants is more time-consuming than existing methods like SAC.